



# Constraints and biases in a tropospheric two-box model of OH

Stijn Naus[1], Stephen A. Montzka[2], Sudhanshu Pandey[3,4], Sourish Basu[2,5], Ed J. Dlugokencky[2], and Maarten Krol[1,3,4]

[1]Meteorology and Air Quality, Wageningen University and Research, the Netherlands
[2]NOAA Earth System Research Laboratory, Global Monitoring Division, Boulder, CO, USA
[3]Institute for Marine and Atmospheric Research, Utrecht University, the Netherlands
[4]Netherlands Institute for Space Research SRON, Utrecht, the Netherlands
[5]Cooperative Institute for Research in Environmental Sciences, University of Colorado, Boulder, CO, USA

**Correspondence:** Stijn Naus (stijn.naus@wur.nl)

**Abstract.**

The hydroxyl radical (OH) is the main atmospheric oxidant and the primary sink of the greenhouse gas $CH_4$. In two recent studies, constraints on the hydroxyl radical (OH) were derived using a tropospheric two-box model of methyl chloroform (MCF) and $CH_4$. When OH variations as derived in this set-up were propagated to the $CH_4$ budget, the constraints on OH from
MCF still allowed for a wide range of $CH_4$ emission scenarios. This is important, because global $CH_4$ emissions are generally considered best constrained by the global lifetime of $CH_4$, which is determined mainly by OH. Here, we investigate how the use of a tropospheric two-box model in these studies can have affected derived constraints on OH, due to the simplifying assumptions inherent to a two-box model. First, instead of prescribing fixed model parameters for interhemispheric transport, chemical loss rates and loss to the stratosphere, we derive species- and time-dependent quantities from a full 3D transport
model simulation. We find significant deviations between the magnitude and time-dependence of the parameters we derive, and the assumptions commonly reported and adopted in literature. Moreover, using output from the 3D model simulations, we investigated differences between the burden seen by the surface measurement network of the National Oceanic and Atmospheric Administration and the true tropospheric burden. Next, we accounted for these biases in a two-box model inversion of MCF and $CH_4$, to investigate the impact of the biases on OH constraints.

We find that the sensitivity of interannual OH anomalies to the biases is modest (1-2%), relative to the significant uncertainties on derived OH (5-8%). However, in an inversion where we implemented all four bias corrections simultaneously, we did find a shift to a positive OH trend over the 1994-2015 period. Moreover, the magnitude of derived global mean OH and by extent that of global $CH_4$ emissions are affected much more strongly by the bias corrections than their anomalies ($\sim$10%). In this way, we identified and quantified direct limitations in the two-box model approach that can possibly be corrected for when
a full 3D simulation is used to inform the two-box model. This derivation is, however, an extensive and species-dependent exercise. Therefore, a good alternative would be to move the inversion problem of OH to a 3D model completely. It is crucial to account for the limitations of two-box models in future attempts to constrain the atmospheric oxidative capacity, especially because though MCF and $CH_4$ behave similarly in large parts of our analysis, it is not obvious that this should be the case for alternative tracers that potentially constrain OH, other than MCF.



# 1 Introduction

Models are a useful tool for the interpretation of measurements. When interpreting atmospheric observations in the context of, for example, atmospheric pollution, or in that of global warming, one often resorts to atmospheric models. Atmospheric models vary in complexity from simple one box models to the most involving 3D transport model. Different types of models are

suitable for addressing different types of problems to different degrees of scrutiny. Therefore, there is no model category that fits all problems. Simple box models are easy to set up, computationally cheap, and transparent. For these and other reasons, their use in atmospheric studies is ubiquitous (e.g. Quay et al. (1999); Walker et al. (2000); Montzka et al. (2011); Schaefer et al. (2016); Schwietzke et al. (2016)). However, simple box models also put limitations on the derived results, as they are by definition less comprehensive than complex models. For example, box models do not explicitly contain much information on a

species' distribution, which can be important if interacting quantities (e.g. loss processes) are distributed non-homogeneously in space. Where exactly these limitations lie and what the gain is from increasing model complexity can be difficult to diagnose, and depends on the application.

A problem that has often been approached in box models is that of constraining the global atmospheric oxidizing capacity, which is largely determined by the hydroxyl radical (OH) (Montzka et al., 2000, 2011). OH is dubbed the detergent of

the atmosphere for its dominant role in the removal of a wide variety of pollutants, including urban pollutants (CO, $NO_x$), greenhouse gases ($CH_4$, HFCs), and HCFCs, which contribute to stratospheric ozone depletion. The budgets of many of these pollutants have been strongly perturbed since pre-industrial times, and it is important to understand what consequences this has had in the past, and could have in the future, for the atmosphere's oxidizing capacity.

Due to its high reactivity, OH has a lifetime of seconds, which inhibits extrapolation of direct measurements. Moreover,

OH abundance is the net effect of many different reactions and reaction cycles, and thus modelling it bottom-up in full-chemistry models is complex and dependent on uncertain emission inventories of the many gases involved. Therefore, the most robust observational constraints on OH on the larger scales are derived indirectly from its effect on tracers: gases that are predominantly removed by OH. Depending on how well the tracer emissions are known, the time evolution of the global mixing ratio of such a tracer can serve to constrain OH. The most well-established tracer for this purpose is methyl chloroform

(MCF) (e.g. Montzka et al. (2000); Bousquet et al. (2005)). In part, this is because it was identified early on as a tracer with a relatively well-defined production inventory (Lovelock (1977); Prinn et al. (1987)). Moreover, production of MCF was phased out in compliance with the Montreal Protocol, and the resulting drop in emissions made loss against OH the dominant term in the MCF budget (Montzka et al., 2011).

Research and debate surrounding OH (Krol and Lelieveld (2003); Krol et al. (2003); Reimann et al. (2005); Prinn et al.

(2005); Rigby et al. (2013)) has lead to considerable improvements in its constraints: most importantly, a likely upper bound on global interannual variability of OH of a few percent (Montzka et al., 2011). Two recent studies derived OH variations in a tropospheric two-box model through an inversion of atmospheric MCF and $CH_4$ observations (Rigby et al. (2017); Turner et al. (2017)). In such an inversion, a range of parameters is optimized (most prominently emissions of MCF and $CH_4$, and OH), so that the modelled mixing ratios best match atmospheric observations of the tracers involved. Both studies found that



constraints on OH in this set-up were weak enough that a wide range of $CH_4$ emission scenarios were still possible. This is an important conclusion, as the $CH_4$ growth rate, combined with the $CH_4$ lifetime (in turn dominated by MCF-derived OH), is generally assumed to provide the strongest top-down constraints on global $CH_4$ emissions and variations therein. We note that in Rigby et al. (2017) the two tropospheric boxes were supplemented by a single stratospheric box, making it technically a
three-box model. However, due to our focus on the troposphere, we hereafter treat this type of model too as a two-box model, and where relevant we discuss the implication of the addition of a stratospheric box.

There are two important reasons to approach the problem of constraining OH in a two-box model. Firstly, through the focus on annual timescales and hemispheric spatial scales, the result is only sensitive to interannual variability in large-scale transport of the modelled tracers. Moreover, by focusing on interannual variability as opposed to absolute OH or emission
levels, remaining systematic offsets do not significantly affect the outcome. The underlying assumption is that either the influence of transport has low inter-annual variability (Turner et al., 2017), or that the important part can be captured by approximating interhemispheric (IH) exchange using $SF_6$ (Rigby et al., 2017). This assumption is necessary, because there is not enough information in the system to constrain all budget terms at once, nor is the full influence of transport explicitly captured in the two-box set-up.

Secondly, a crucial part of the optimization consists of disentangling the influence of OH and that of emission variations on observed MCF mixing ratios. Ideally, MCF emission variations would be prior knowledge. However, though MCF production is well documented, the emission timing is much more uncertain (McCulloch and Midgley, 2001). MCF was mainly used as a solvent in, for example, paint and degreasers of metals. In these applications, MCF is released only when used, rather than when produced, which results in uncertainty in the emission timing. Moreover, due to the continuing decline of the atmospheric MCF
mixing ratios, small, ongoing MCF emissions could eventually become important. Emissions exceeding bottom-up emission inventories have been identified both from the U.S. (Millet and Goldstein, 2004) and from Europe (Krol et al., 2003), as well as from natural processes, such as MCF re-release from the ocean (Wennberg et al., 2004). Therefore, in the absence of other constraints, emission uncertainties would strongly limit the use of MCF for deriving interannual variability of OH. However, in a two-box set-up, an additional constraint is provided by the IH gradient of MCF. Emission inventories show that MCF
emissions are predominantly located in the Northern Hemisphere (NH), whereas OH has a NH to SH ratio that is uncertain, but the ratio has a likely range of 0.85-1.15 (Patra et al., 2014). Therefore, these two processes have a very different effect on the IH gradient of MCF: a parameter implicitly optimized in a two-box model. Thus, the IH gradient is an important piece of information that can help to disentangle the influence of emissions from the influence of OH on MCF growth rate variations. This use of the IH gradient for constraining global emissions of anthropogenically emitted gases has also been recognized in
previous research (Liang et al. (2017); Montzka et al. (2018)).

Despite the appealing degree of simplicity offered by the two-box model, its results still hinge on many simplifying assumptions, both explicit (e.g. interhemispheric transport) and implicit (e.g. intrahemispheric transport). In this context, the uncertain outcomes of the two recent two-box model studies put forward an important question: how do the simplifying assumptions inherent to the two-box set-up affect the conclusions drawn from it? Or, reversely, would these conclusions change when moving
the analysis to a 3D transport model? A recent study (Liang et al., 2017) partly explored these questions. The study investigated





how to incorporate information from 3D transport models in a two-box model, to increase the robustness of two-box model derived constraints on OH. They found that there are key parameters in the two-box model that can be tuned to better represent the 3D simulation results, and thus ideally better represent atmospheric transport in general. For example, they found that IH transport rates can be strongly species-dependent.

Here, we provide a different approach to the issue. In the first part of our study, we parametrized results from the 3D global transport and chemistry model TM5 into a two-box model. Through this parametrization, we explored difficulties in the translation from the 'reality' of a 3D transport model to a two-box model, and the assumptions made in the process. We focus on four aspects of the parametrization. Firstly, in previous research that involved 3D transport models, three two-box model parameters have been indicated to behave differently than commonly assumed. The tracer-dependent nature of IH transport found by

Liang et al. (2017) was mentioned already: a dependence generally not accounted for in two-box modelling studies. The second parameter is the IH OH ratio. Previous research has shown that because of tracer-specific source-sink distributions, different tracers can be exposed to different global mean OH concentrations (Lawrence et al., 2001). Here, we extend this result to a species- and time-dependent IH OH ratio. Additionally, for MCF specifically, net loss to the stratosphere might be slowing after its emissions dropped (Krol and Lelieveld (2003); Bousquet et al. (2005)). We will resolve this issue by deriving the relevant

stratospheric loss parameter from full 3D simulations. Finally, we used the 3D simulation to investigate differences between the burden seen by the surface measurement network of the National Oceanic and Atmospheric Administration (NOAA) and the true tropospheric and hemispheric burden in our 3D model: a bias that was also explored by Liang et al. (2017). In the second part of this study, we assess the impact of these four potential biases on derived OH variations in a two-box inversion set-up that is very similar to Rigby et al. (2017) and Turner et al. (2017). The objective is to provide a quantitative estimate

of the impact of biases in a two-box inversion, and to explore if and how these can be accounted for. Though this study is focused on the problem of OH, it also serves as a case study of potential pitfalls in two-box models in general, when applied to interpreting global-scale atmospheric observations.

## 2   Methods

### 2.1   Two-box inversion

In this section, we discuss the set-up of our two-box model inversion. The model incorporates two tracers (MCF and $CH_4$) and consists of two boxes (the troposphere in the NH and in the SH), which are delineated by a fixed equator. The stratosphere is implicitly included in the model through a first-order loss process that is taken to be equal for both hemispheres. The governing equations for a tracer mixing ratio $X$ are given in Equation 1.

$$\frac{dX_{NH}}{dt} = E_{NH} - (k_{OH}[OH_{NH}] + l_{strat} + l_{other})X_{NH} - k_{IH}(X_{NH} - X_{SH}), \tag{1a}$$

$$\frac{dX_{SH}}{dt} = E_{SH} - (k_{OH}[OH_{SH}] + l_{strat} + l_{other})X_{SH} + k_{IH}(X_{NH} - X_{SH}). \tag{1b}$$





Thus, within each hemisphere, there are emissions ($E$), loss to OH ($k_{OH}[OH]X$), loss to the stratosphere ($l_{strat}X$), loss to other processes ($l_{other}X$; e.g. loss to the ocean), and transport between the hemispheres ($k_{IH}(X_{NH} - X_{SH})$). The model runs at an annual timestep. The fundamentals of this model set-up are also found in Rigby et al. (2017) and Turner et al.

(2017), though the exact treatment of the different budget terms can differ. For example, Turner et al. (2017) combined all tropospheric loss, including loss to the stratosphere, in one term, whereas Rigby et al. (2017) included a stratospheric box, so that stratospheric loss becomes a transport rather than a first-order loss term. Where relevant, we will point out further differences with these previous studies.

Since the objective is to leverage observed mixing ratios to infer information on tropospheric OH, we also need an inverse

estimation framework, complementary to the above forward model. The objective of the inversion is to optimize a state $x$, such that the forward model best reproduces the observations, without straying too far from a first best guess: the prior. Therefore, the state is the vector which contains all parameters that need to be optimized. The optimization objective is analogous to minimizing the cost function $J$, as defined in Equation 2:

$$J(\boldsymbol{x}) = \frac{1}{2}(\boldsymbol{x} - \boldsymbol{x}_{prior})^T \mathbf{B}^{-1}(\boldsymbol{x} - \boldsymbol{x}_{prior}) + \frac{1}{2}(\mathbf{H}\boldsymbol{x} - \boldsymbol{y})^T \mathbf{R}^{-1}(\mathbf{H}\boldsymbol{x} - \boldsymbol{y}),\qquad(2)$$

with $\mathbf{B}$ and $\mathbf{R}$ the prior and observation error covariance matrix respectively, $\mathbf{H}$ the forward model, and $\mathbf{y}$ the observations. In addition, we compute the cost function gradient $\nabla J$ (Equation 3).

$$\nabla J(\boldsymbol{x}) = \mathbf{B}^{-1}(\boldsymbol{x} - \boldsymbol{x}_{pri}) + \mathbf{H^T}\mathbf{R}^{-1}(\mathbf{H}\boldsymbol{x} - \boldsymbol{y}),\qquad(3)$$

with $\mathbf{H^T}$ the transpose of the forward model, also known as the adjoint model. Note that because the forward model $\mathbf{H}$ is non-linear (e.g. OH chemistry), we use the adjoint of the tangent-linear forward model. Calculation of the cost function

gradient facilitates quicker convergence of the optimization. For the minimization we use the Broyden-Fletcher-Goldfarb-Shanno algorithm. In essence, this statistical inversion set-up is the same as that used in the 4DVAR system of ECMWF (Fisher, 1995) and TM5-4DVAR (Meirink et al., 2008).

For the optimization of MCF emissions, we use an extended version of the emission model from McCulloch and Midgley (2001). This emission model is adopted to account for the varying and uncertain release rates of MCF when used in different

applications (e.g. degreasing agent, paint). This results in a gap between the uncertainty in production, or integrated emissions (~2%), and the uncertainty in annual emissions (up to 40%) (McCulloch and Midgley, 2001). Therefore, production is distributed between four different categories with different release rates: rapid, medium, slow and stockpile. In the prior distribution, the bulk of production ($> 95\%$) is placed in the rapid category. To account for uncertainty in the production inventory, we also adopt an additional emission term superimposed on the production-derived emissions. The emissions in year

$i$ are then given by Equations 4 and 5. For each year $i$, we optimize four parameters for MCF emissions: three parameters that shift emissions between the rapid production category and each of the other three categories ($f^i_{Medium}$, $f^i_{Slow}$ and $f^i_{Stock}$ in




Equation 6), and the additional emissions term ($E^i_{Additional}$), which has an uncertainty constant through time. This emission model is similar to that used in (Rigby et al., 2017), though ours leaves more freedom to emission timing.

$$E^i = E^i_{Rap} + E^i_{Med} + E^i_{Slow} + E^i_{Stock} + E^i_{Additional},\qquad(4)$$

for the emissions in year $i$, where:

$$
\begin{aligned}
E^i_{Rap} &= 0.75 P^i_{Rap} + 0.25 P^{i-1}_{Rap},\\
E^i_{Med} &= 0.25 P^i_{Med} + 0.75 P^{i-1}_{Med},\\
E^i_{Slow} &= 0.25 P^{i-1}_{Slow} + 0.75 P^{i-2}_{Slow},\\
E^i_{Stock} &= \sum_{j=1}^{11} P^{i-j}_{Stock},
\end{aligned}
\qquad(5)
$$

and, in the optimization:

$$
\begin{aligned}
P^i_{Rap} &= (1 - f^i_{Med} - f^i_{Slow} - f^i_{Stock}) P^i_{Rap,prior},\\
P^i_{Med} &= P^i_{Med,prior} + f^i_{Med} P^i_{Rap,prior},\\
P^i_{Slow} &= P^i_{Slow,prior} + f^i_{Slow} P^i_{Rap,prior},\\
P^i_{Stock} &= P^i_{Stock,prior} + f^i_{Stock} P^i_{Rap,prior}.
\end{aligned}
\qquad(6)
$$

An important choice in the inversion set-up is which parameters to prescribe and which to optimize. Rigby et al. (2017) optimized all parameters, so as to explore the full uncertainty of the optimization within the inversion framework. Turner et al. (2017) only optimized hemispheric MCF and CH$_4$ emissions and hemispheric OH, while the remaining uncertainties were partly explored in sensitivity tests. We choose to optimize four end-products for each year: global OH, global MCF emissions, global CH$_4$ emissions, and the CH$_4$ emission fraction in the NH. Thus we have a closed system, as we also fit to four observations: the global mean mixing ratio and the IH gradient of both MCF and CH$_4$. In addition to the 4DVAR inversion, we generate a Monte Carlo ensemble, where in each realization the prior and the observations are perturbed, relative to their respective uncertainties. Then, the new prior is optimized using the new observations. The Monte Carlo simulation quantifies the sensitivity of the optimization to the prior choice and to the realization of the observations. The Monte Carlo set-up also allows us to explore the sensitivity of the inversion to parameters that are not optimized, such as the fraction of MCF emissions in the NH. This approach has the added advantage that parameters that are perturbed in the Monte Carlo simulation, but not optimized in the 4DVAR system, do not need to have a Gaussian error distribution, which would normally be a prerequisite in a 4DVAR inversion. The specifics of our inversion set-up are given in Table 1.



**Parameters optimized in inversion**

**and perturbed in Monte Carlo (Gaussian)**

| Parameter | Prior estimate | Uncertainty |
|---|---|---|
| Global MCF emissions | Based on McCulloch and Midgley (2001) | |
| - $f_{Medium}$ | 0% | 5% |
| - $f_{Slow}$ | 0% | 5% |
| - $f_{Stock}$ | 0% | 5% |
| - Unreported emissions | 0 Gg/yr | 10 Gg/yr |
| Global CH$_4$ emissions | 550 Tg/yr | 15% |
| Global OH | $9 \times 10^5$ molec/cm$^3$ | 10% |
| Fraction NH CH$_4$ emissions | 75% | 10% |

**Parameters not optimized in inversion,**

**but perturbed in Monte Carlo (Uniform)**

| Parameter | Lower bound | Upper bound |
|---|---|---|
| Fraction NH MCF emissions | 90% | 100% |

**Parameters not optimized in inversion**

**and not perturbed in Monte Carlo**

| Parameter | Standard | Alternative |
|---|---|---|
| Interhemispheric OH ratio | 0.98 | TM5-derived∗ |
| MCF lifetime w.r.t. oceanic loss | 83 yr | - |
| MCF lifetime w.r.t. stratospheric loss | 45 yr | TM5-derived∗ |
| CH$_4$ lifetime w.r.t. stratospheric loss | 150 yr | TM5-derived∗ |
| Interhemispheric transport | 1 yr$^{-1}$ | TM5-derived∗ |

∗See Section 2.2

**Table 1.** The relevant settings we use in the inversion of our two-box model. The upper section contains the parameters optimized in the inversion, which are also perturbed in the Monte-Carlo ensemble. These parameters have Gaussian uncertainties, and their mean and 1-$\sigma$ uncertainty are given. The middle section contains parameters that are perturbed in the Monte Carlo, but not optimized. The middle parameters have uniform uncertainties, of which the lower and upper bound are given. The bottom section contains parameters that are neither optimized nor perturbed. For these parameters, the left column gives the standard setting, whereas the alternative column indicates whether we also ran an inversion using a TM5-derived timeseries (see Section 2.2.2).



### 2.2 TM5 set-up and two-box parametrizations

#### 2.2.1 3D model set-up

For the 3D model simulations we use the atmospheric transport model TM5 (Krol et al., 2005). The model is operated at a $6° \times 4°$ horizontal resolution, at 25 vertical hybrid sigma pressure levels. The analysis period is 1988-2015, where we treat
1988 and 1989 as spin-up years. TM5 transport is driven by meteorological fields from the ECMWF ERA-Interim reanalysis (Dee et al., 2011). Convection of tracer mass is based on the entrainment and detrainment rates from the ERA-Interim dataset. This is an update from the previous convective parametrization used by, for example, Patra et al. (2011). The new convective scheme results in in faster inter-hemispheric exchange of tracer mass, more in line with observations (Tsuruta et al., 2016).

We run TM5 with three tracers: $CH_4$, MCF and $SF_6$. For $CH_4$, we annually repeat the 2009-2010 a priori emission fields
used by Pandey et al. (2016), and we also use the same fields for stratospheric loss to Cl and $O(^1D)$. For MCF, we use emissions from the TransCom-CH4 project (Patra et al., 2011). Since these emissions are available only up to 2006, we assume a globally uniform exponential decay of 20%/year afterwards, similar to Montzka et al. (2011). MCF-specific loss fields (ocean deposition and stratospheric photolysis) are also taken from the TransCom-CH4 project. Details of the MCF loss and emission fields can be found in the TransCom-$CH_4$ protocol. The OH loss fields we use are a combination of the 3D fields from Spivakovsky et al.
(2000) in the troposphere and stratospheric OH as derived using the 2D MPIC chemistry model (Brühl and Crutzen, 1993). The OH fields are scaled by a factor 0.92, as described by Huijnen et al. (2010). For $SF_6$, we use emission fields from the TransCom Age of Air project (Krol et al., 2017), with no loss process implemented.

Since the above set-up is simplistic in some aspects (e.g. annually repeating $CH_4$ emissions), we also ran a 'nudged' simulation, which is discussed in Supplement 2. The nudged simulation provides a test of the sensitivity of our results to the
source-sink distributions we use in the 3D simulation. In general, we find that our results are relatively insensitive to nudging.

#### 2.2.2 Parametrizing 3D model output to two-box model input

We use the TM5 simulations to inform the two-box model, by reducing the 3D model output to two-box model parametrizations. The objective of this parametrization is to derive parameters that can be used in the two-box model defined in Equation 1, such that this budget is closed. To parametrize the 3D output of TM5 into our two-box model, the 3D fields were divided in
3 boxes: the troposphere in the NH and in the SH, and the stratosphere. The border between the hemispheres is taken as the equator (i.e. fixed in time). Where relevant we discuss the sensitivity of our results to this demarcation. We define a dynamical tropopause as the lowest altitude where the vertical temperature (T) gradient is smaller than 2 K/km, clipped at a geopotential height of 9 and 18 km. Our analysis was found to be insensitive to the exact definition of the tropopause. Next, we computed an annual budget for each box. For the two tropospheric boxes, this was done as in Equation 1. This was supplemented by
Equation 7 for the stratospheric box.

$$\frac{dX^{Strat}}{dt} = -L^{Strat}_{local} + l_{strat}(X^{SH} + X^{NH}),$$
(7)



Emissions, local loss and mixing ratios per box can be derived from the 3D model in- and output, and thus $l_{strat}$ and $k_{IH}$ can be inferred from these equations. Note that we have do not strictly need the stratospheric budget equation to resolve two parameters, but we use it to resolve numerical inaccuracies. Resolving the budget of each species in this manner provides the necessary input of the tropospheric two-box model defined in Section 2.1, such that on the hemispheric and annual scale,
identical results are obtained with the 3D and the two-box model.

### 2.2.3    Model-sampled observations

To assess the quality of the observation-derived hemispheric burden with respect to the calculated hemispheric burden in the TM5 model, we generated a timeseries of observations by subsampling TM5 output at NOAA sampling sites, at the same sampling times: hereafter referred to as model-sampled observations. The standard in tracking global trends in atmospheric
trace gases are surface measurement networks: for $CH_4$ and MCF most notably the NOAA (Dlugokencky et al. (2009); Montzka et al. (2011)) and AGAGE (Advanced Global Atmospheric Gases Experiment) (Prinn et al., 2018) networks. By selecting measurement sites far removed from sources, the theory is that a small number of sites already puts strong constraints on the global growth rate (Dlugokencky et al., 1994). In general, quantification of the robustness of the derived growth rates based solely on observations can be difficult, since there are likely systematic biases inherent to sampling a small number of surface
sites. When assimilated into a 3D transport model, these biases will largely be resolved (if transport is correctly simulated). However, when the data is aggregated to two hemispheric averages, as in a two-box model, quantification of the potential biases is crucial.

     Here, we explore the resulting bias in our model framework. By subsampling the TM5 output at the locations of NOAA stations, at NOAA measurement instances, we generate a set of model-sampled observations. These model-sampled observations
are intended to be as representative as possible for the real-world observations of the NOAA network. To aggregate the station data to hemispheric averages, we use methods similar to those deployed by NOAA (for MCF: Montzka et al. (2011), with further details on our adaption in Supplement 1; for $CH_4$: Dlugokencky et al. (1994)). By comparison of the resulting products with the calculated tropospheric burden, as derived from the full tropospheric mixing ratios, we can assess how well the burden derived from the NOAA network represents the model-simulated tropospheric burden. The two end-products we investigate
for each tracer are the rate of change of the global mean mixing ratio and that of the IH gradient. Note that by mixing ratio we mean the dry air mole fraction. These two parameters best reflect the information as it is used in a two-box model: the global mean mixing ratio is used to constrain the combined effect of OH and emissions, while the IH gradient is used to distinguish between the two. Note that in previous box-model studies of MCF, often only global growth rates were derived (Montzka et al., 2000, 2011).

### 2.3    Potential biases in the two-box model

By concentrating on the budget of MCF, we identified three parameters that deviated significantly from what is generally expected when using a two-box model to constrain OH. In addition, we investigated the potential bias in converting station



data to hemispheric averages (see Section 2.2.3 and Supplement 1). We will quantify these biases and propagate them in two-box model inversions in Section 3.2 to understand their impact on derived quantities related to OH.

### Interhemispheric transport

IH transport is a key parameter in the two-box model. IH transport of tracer mass can vary because of variations in IH transport of air mass (e.g. influenced by ENSO, particularly at Earth's surface (Prinn et al. (1992); Francey and Frederiksen (2016); Pandey et al. (2017))), or because of variations in the source-sink distribution, and thus of the tracer's concentration distribution itself. Generally, interannual variability in IH transport is considered to be in the order of 10% (Patra et al., 2011). Two-box models tend to assume time-constant IH exchange (Turner et al., 2017) and/or similar exchange rates for different

tracers (Rigby et al., 2017). However, it is uncertain whether such assumptions hold for a tracer which undergoes strong source-sink redistributions, such as MCF. The IH transport variations we derived for each tracer are discussed in Section 3.1.1.

### Surface sampling bias

As discussed in Section 2.2.3, we explored the bias that results from representing hemispheric averages using sparse surface

observations. Surface networks are a valuable resource, because they provide high-quality, long-term measurements of a growing variety of tracers. However, temporal and horizontal coverage of surface networks is limited. For example, coverage in the tropics, where latitudinal gradients tend to be highest and most variable, is sparse or absent. Moreover, surface measurements do not inform much on vertical gradients. More vertical information from observations is available in the form of surface sites at elevation (e.g. the observatories at Summit, Greenland and Mauna Loa, Hawaii) and aircraft campaigns (e.g. the HIPPO

campaign (Wofsy et al., 2011)), but this information is difficult to correctly incorporate in a hemispheric average. These are important shortcomings when using the surface networks as input for a two-box model of the troposphere. In Section 3.1.2 we discuss how these limitations can result in biases in two-box model observations.

### The interhemispheric OH ratio

The IH ratio of OH concentrations is an uncertain parameter. This is mostly because of a mismatch between results from full-chemistry models (1.13-1.42 (Naik et al., 2013)) and from MCF-derived constraints (0.87-1.07 Patra et al. (2014)). The latter is generally the loss ratio considered in two-box models (1.0 in Turner et al. (2017) and 0.95-1.20 in Rigby et al. (2017)), and is similar to the ratio we used in the TM5 simulations (0.98 (Spivakovsky et al., 2000)). However, the bias we consider here is of a different nature: it is the difference between the physical IH OH ratio and the IH ratio a particular tracer is exposed to.

It is known that different tracers can be exposed to different oxidative capacities (Lawrence et al., 2001). Therefore, different tracers might similarly be influenced by different IH ratios in OH, which is mostly driven by variations in a tracer's respective source/sink distributions. This bias we explore in Section 3.1.3.

### MCF loss to the stratosphere




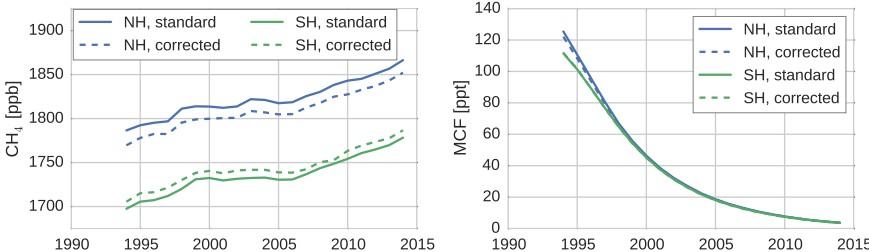

**Figure 1.** Hemispheric, annual mean timeseries of $CH_4$ (left) and MCF (right), as derived from the NOAA surface sampling network. Solid lines denote averages as derived directly from the NOAA surface sampling network (which are used in our standard inversion). Dashed lines denote the same timeseries, but adjusted by correction factors that were derived from our TM5 simulations. Figure 3 shows the ratios between the standard and corrected timeseries.

The second-most important loss process of MCF is stratospheric photolysis. In our TM5 set-up, this loss process results in an in-stratosphere lifetime (stratospheric burden/stratospheric loss) of 4 to 5 years. It is generally assumed that this in-stratosphere loss translates to a lifetime of global MCF with respect to the stratosphere (global burden/stratospheric loss) of 40 to 50 years (Naik et al. (2000); Ko et al. (2013)), which corresponds to ~10% of global MCF loss. Rigby et al. (2017) assumed a constant

stratospheric lifetime of MCF of 40 (29−63) year, while Turner et al. (2017) incorporated this loss process in the OH loss term. Due to the rapid drop in MCF emissions and the relatively slow nature of troposphere-stratosphere exchange, however, this lifetime could vary through time (Krol and Lelieveld (2003); Bousquet et al. (2005)). We will investigate this in Section 3.1.4.

### 2.4 Standard two-box inversion and bias correction

To assess the impact of the biases discussed in Section 2.3 on a two-box model inversion, we run our inversion (see Section 2.1)
using different settings. In the standard, default inversion we do not consider any of the four biases discussed above. Thus, we use constant IH exchange (1 year), constant stratospheric loss of MCF (45 years), and a constant IH OH ratio (0.98) (see Table 1). The first three potential bias corrections are then straightforwardly implemented by replacing these constant values with the timeseries we derived for each parameter from the full 3D simulations. As mentioned in Section 2.1, we do not include uncertainties in these three parameters, as this would obscure the impact of the bias correction. For the surface sampling bias, we
first compute a correction between the hemispheric means as derived from the model-sampled observations and the calculated (TM5) hemispheric, tropospheric means (with demarcation at the equator). Then, we apply this correction to the real-world NOAA hemispheric means we use in the standard inversion. This gives a new set of observations, which we use in the inversion (discussed in Section 3.1.2). Both the standard and the corrected set of observations are shown in Figure 1. Through comparison of the outcome of the standard inversion and an inversion with one or more biases implemented, we can evaluate the individual
and cumulative impact of the biases on derived OH and $CH_4$ emissions.





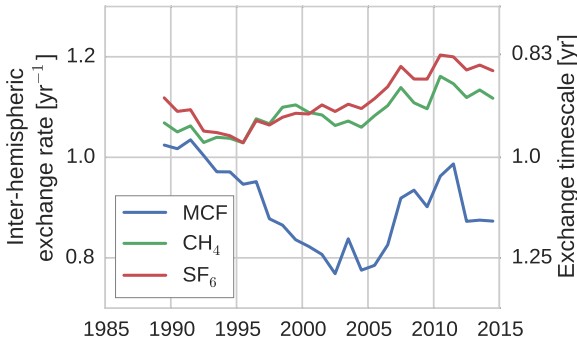

**Figure 2.** The IH exchange rate for MCF, CH$_4$ and SF$_6$, as derived from a two-box parametrization of TM5 output.

## 3  Results

### 3.1  Biases

#### 3.1.1  Interhemispheric transport

The IH exchange coefficients, derived for the three different tracers from Equations 1 and 7, are shown in Figure 2. Clearly, the
exchange rates differ both in mean value, as well as in interannual variability. MCF is the clear outlier, but SF$_6$ and CH$_4$ also
show different variations. The drivers of these differences are differences in intrahemispheric tracer distributions and in the
underlying source and sink distributions. The three tracers differ strongly in this respect: SF$_6$ and MCF only have emissions
in the NH mid-latitudes, whereas CH$_4$ has significant emissions in the tropics and in the SH. SF$_6$ has no sink, whereas MCF
and CH$_4$ have a sink with a distinct tropical maximum in OH. This all affects how IH transport of air mass translates to IH
transport of tracer mass, or the derived IH transport rate as derived from a full 3D model with interannually-varying transport.

Most notable is the minimum in the IH exchange rate for MCF in the 2000-2005 period. The timing of the 1989-2003 decline
in $k_{IH}$ coincides with the initial drop in MCF emissions. An important shift in the distribution of the MCF mixing ratio is that
the global minimum shifts from the South Pole to the tropics. Thus, the latitudinal gradient is no longer unidirectional, but
changes signs in the tropics. In the same period, there is a strong vertical redistribution which has also likely impacted IH
exchange. It is not obvious that these changes should result in slower IH exchange, but in the end, in TM5, they do.

Another notable feature is the positive trend in the IH exchange rate for CH$_4$ (+0.35 $\pm$ 0.05 %/yr; p=0.00) and for SF$_6$
(+0.50$\pm$ 0.01 %/yr; p=0.00). For CH$_4$, we use annually repeating sources, so that changes in the source-sink distribution are
unlikely to contribute to the trend or to the variability. Indeed, in a simulation with annually repeating meteorology, we find
near-zero variability in $k_{IH}$ for CH$_4$ (results not shown). Therefore, there is something in the combination of the meteorological
data, the treatment of this data in TM5 and the source-sink distribution of both CH$_4$ and SF$_6$ which results in a significantly
positive trend in the IH exchange rate of both gases. This trend could either indicate an acceleration of IH transport of air mass,



or a shift in the pattern of IH transport which favours IH exchange of $CH_4$ and $SF_6$. It is unclear from this analysis what the underlying mechanism is exactly, except that it is driven by temporal variations in transport and thus that there are parameters in the meteorological fields which also show a trend: otherwise this final product cannot exhibit a trend. However, it might be that the sensitivity of TM5 transport to these parameters is biased.

We also investigated this trend in the results of the TransCom Age of Air experiment (Krol et al., 2017). We looked at the age at the South Pole surface of a tracer which is forced uniformly at the NH surface, according to the AoA protocol (Krol et al., 2017). For both TM5 run at a $3° \times 2°$ resolution and for the LMDZ5A model, nudged to the ECWMF ERA interim meteorology, (Hourdin et al., 2013) we find a negative trend of $\sim 0.2\%$/year over the 1995-2014 period. This trend persists even if we sample the same tracer in the free troposphere at SH mid-latitudes. Younger NH air at the South Pole indicates

an increase in the IH exchange rate, and the magnitude of the trend is similar to what we have identified from Figure 2. The agreement between the two models suggests some robustness of the trend to different transport parametrizations and to model resolution. Though interesting, it is beyond the scope of this paper to investigate this issue further.

  To test the sensitivity of the derived IH exchange rates to the source-sink distribution, we compared $k_{IH}$ derived from the standard simulation to the nudged simulation (the nudging procedure is explained in Supplement 2). IH transport of $CH_4$ as

derived from the nudged simulation showed higher interannual variations than in the standard simulation (results not shown), which can be expected, as the source-sink distribution becomes more variable. However, the general characteristics were conserved: most notably, the positive trend over the entire period persists, for $CH_4$ and for $SF_6$. For MCF, we find that the general characteristics of derived $k_{IH}$ are similarly insensitive to nudging, with the main change being a deeper 2000-2005 minimum in the nudged simulation. In the end, we deem the anomalies presented in Figure 2 quite robust and relatively

insensitive to the exact spatio-temporal source-sink distribution.

  When the hemispheric interface is shifted from the equator to $8°N$, which is more representative of the average position of the ITCZ, the IH exchange rate increases for all tracers, but the variability in IH exchange of $CH_4$ and $SF_6$ remains largely unaffected. However, for MCF, the variability shifts completely. Rather than decreasing after the emission drop, the IH exchange rate now increases. This sensitivity reflects that for a tracer with a relatively small IH gradient which minimizes in the tropics,

it becomes difficult to define an IH transport rate in a two-box model. By extension, care should be taken when interpreting the IH gradient of MCF in later years, since the influence of IH transport is difficult to isolate.

### 3.1.2 Surface sampling bias

Figure 3 shows the surface network bias in the global mean mixing ratios and in the IH gradient. The bias is quantified as the ratio between values derived from the model-sampled observations (see Section 2.2.3) and values derived from the full

(TM5) troposphere. A comparison with global mean mixing ratios derived from real-world NOAA observations is given in Supplement 2.

  The bias in the IH gradient is particularly large, because averages based on NOAA surface stations systematically overestimate the tropospheric burden in the NH and underestimate the burden in the SH. There are two important effects contributing to this bias. Firstly, in the NH, where most emissions are located, mixing ratios tend to decrease with altitude, while in the SH



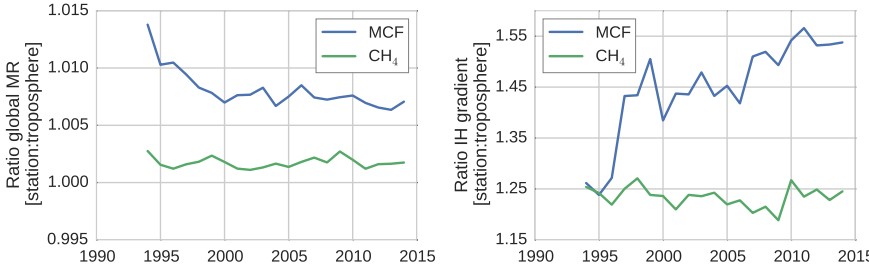

**Figure 3.** The surface sampling bias in the global mixing ratio (left) and in the IH gradient (right) of MCF and of $CH_4$. The bias is quantified as the ratio between values derived from the NOAA surface sampling network and values derived from the full (TM5) troposphere. Figure 1 visualizes the impact of correcting for the sampling bias in real-world NOAA observations.

|  | **Global** | **IH gradient** |
|---|---|---|
|  | **growth rate** | **rate of change** |
| **$CH_4$** | 0.96 ppb/yr / 0.05 %/yr | 2.56 ppb/yr / 0.13 %/yr |
| **MCF** | - / 0.14 %/yr | - / 0.33 %/yr |

**Table 2.** Mean observational errors as derived from TM5 simulations over the 1994-2015 period. The errors are quantified as the mean difference between annual means derived from model-sampled observations and annual means derived from the full tropospheric grid. $CH_4$ uncertainties are given both in ppb/yr and relative to the global mean mixing ratio. Uncertainties for MCF are only given relative to the global mean, because of its strong temporal decline.

vertical gradients are much smaller, or even reversed. Secondly, latitudinal gradients of both MCF and $CH_4$ tend to be highest in the tropics, where few or no measurement sites are available. Again, due to high emissions in the NH, mixing ratios in the NH decrease towards the equator, while mixing ratios increase towards the equator in the SH. Both biases are of opposite sign in each hemisphere. Thus, in a global average, these biases largely cancel, and only a small overestimate remains (left panel in Figure 3). For the IH gradient, however, these biases add up, which results in an overestimate of the IH gradient by surface stations of up to 20-40% (right panel in Figure 3). For MCF before 1995 and for $CH_4$ throughout the analysis period, the bias from the vertical gradient dominates. The shift in the bias for MCF is driven by the latitudinal dimension. The MCF gradient gets a minimum in the tropics and apparently this exacerbates the effect of the lack of tropical stations, combined with simple latitudinal interpolation.

We note that the derived bias in the IH gradient is sensitive to the demarcation of the two tropospheric boxes. If we shift the IH interface from the equator to 8°N, the bias is reduced to 15% for $CH_4$, and varies between 15 and 25% for MCF. The trend in the IH bias of MCF becomes smaller, but persists.



Liang et al. (2017) performed a similar analysis. They report a similar low to absent bias in the global mean and a more significant bias in the IH gradient ($\sim 10\%$). This is smaller than the bias we find, even if we demarcate the hemisphere at $8°$N. However, an important difference is that in Liang et al. (2017) model-sampled observations are compared to the surface grid, instead of to the full troposphere. Thus, their bias estimate does not include vertical effects. If we use the surface grid as a

reference, the IH bias for $CH_4$ is reduced to $-10\%$: i.e. it reverses. This indicates that sampling the background atmosphere results in an underestimate of the surface IH gradient. For MCF the bias shift persists, and the maximum bias is only slightly reduced to 15%, indicating a dominant influence from the latitudinal dimension. We emphasize that for a tropospheric two-box model the comparison with the full troposphere is most relevant.

This analysis also provides an estimate of uncertainties in the rate of change of the global mixing ratio and in that of the

IH gradient: the relevant observational parameters in a two-box inversion. Table 2 gives the differences between the quantities derived from model-sampled observations and from the full troposphere, i.e. the "true" (TM5) error. We can compare this TM5-derived uncertainty to uncertainties derived only from observations, which we use in the two-box inversions. For $CH_4$, we use uncertainties as reported by NOAA. These are obtained by generating an ensemble of surface network realizations, where in each realization different sites are excluded or double-counted randomly (bootstrapping). For each realization, aggregated

quantities such as the global mean growth rate can be derived. The spread within the ensemble then provides a measure for the uncertainty. For MCF no such uncertainties are reported. Therefore, we developed our own method, which is described in Supplement 1.

Observation-derived uncertainties in the global mean growth rate are around 0.60 ppb/yr and 0.6%/yr for $CH_4$ and for MCF respectively. NOAA does not explicitly report an uncertainty in the IH gradient of $CH_4$, but error propagation from hemispheric

means gives an uncertainty of 1.1 ppb/yr. For MCF, we find a time-dependent uncertainty in the rate of change of the IH gradient of $1.0 - 1.5\%$.

The $CH_4$ errors we derive from the TM5 simulation are slightly higher than the uncertainties reported by NOAA. Furthermore, since we use annually repeating $CH_4$ emissions, variations in $CH_4$ emissions can further increase the error. Indeed, the nudged run (see Supplement 2) results in 20% higher uncertainties. However, is important to note that the $CH_4$ uncertainties re-

ported by NOAA are intended to reflect the match with the marine boundary layer (MBL), rather than with the full troposphere. Therefore, it is not surprising that the errors we find are somewhat higher.

For MCF, we adopt observation-derived uncertainties that are significantly lower than those used by Rigby et al. (2017) and Turner et al. (2017): both studies report uncertainties of around 5% in hemispheric averages. Both studies use different methods, that are grounded on different observational information. In Rigby et al. (2017), temporal variability dominates the uncertainty

estimate, while in Turner et al. (2017) spatial variations are used. Our method is more similar to Rigby et al. (2017), but with modifications that average out some of the temporal variability, under the assumption that variability at different measurement sites is largely uncorrelated (details in Supplement 1). All three methods are defensible, and other defensible methods resulting in yet different estimates are also conceivable. This serves to show that observation-derived uncertainties in MCF averages are uncertain quantities, in large part due to the relatively low number of available surface sites. Therefore, the uncertainty derived

from TM5 is an especially useful addition for MCF.





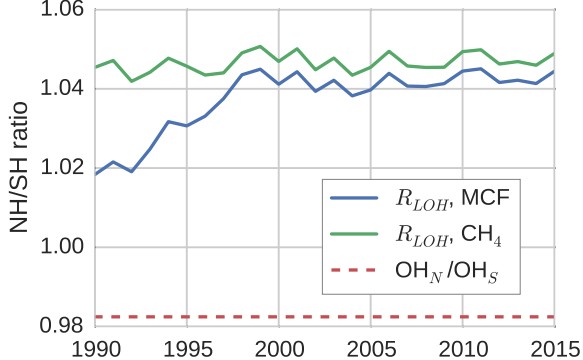

**Figure 4.** The ratio between tracer lifetime with respect to OH loss in the NH troposphere and SH troposphere. Additionally, the IH ratio in OH concentrations is shown.

Here we see that TM5-derived uncertainties in MCF averages are significantly lower than our already low observation-derived estimates. It indicates that even the use of a simple averaging algorithm and a small number of surface sites, relative to what is available for $CH_4$, already results in well-constrained hemispheric and global growth rates. The TM5-derived estimate thus supports the use of our observation-derived uncertainty estimates, rather than the higher estimates used in previous studies.

Our uncertainty estimates, as well as the NOAA uncertainty estimates, do not include measurement uncertainty. On a hemispheric, annual scale the influence of random measurement uncertainty is insignificant, but scale drifts could affect our estimates: for example, Rigby et al. (2013) reported same-site differences between NOAA and AGAGE observations of MCF that drifted in a range of 0 to 2% over a 20-year period.

### 3.1.3    Interhemispheric OH ratio

In the TM5 simulations from which the global loss rates are derived, the prescribed tropospheric OH fields were taken from Spivakovsky et al. (2000). In these fields, the IH OH ratio is 0.98, when the IH interface is considered to be the equator. One might expect a similar ratio between OH loss in the NH and in the SH, which we quantified through the ratio between hemispheric lifetimes with respect to OH ($\tau_{OH}$) in Equation 8. We find that this is not the case (see Figure 4).

$$r_{L_{OH}} = \frac{\tau_{X,OH}^{NH}}{\tau_{X,OH}^{SH}}. \tag{8}$$

The loss ratio is up to 7% higher than the physical OH ratio. Moreover, the ratio is not the same for MCF and $CH_4$, and the ratio that corresponds to MCF shows a trend. We found that the systematic positive offset is largely driven by an IH asymmetry in the spatio-temporal correlations between OH and temperature. Mostly, this is because the OH maximum in the NH is located at lower altitude than in the SH. Since at low altitudes temperatures are higher, and higher temperatures correspond to higher





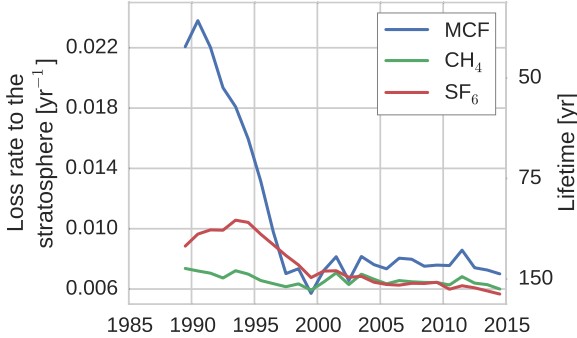

**Figure 5.** The tropospheric loss rate to the stratosphere, as derived from the TM5 simulations.

reaction rates, this asymmetry results in relatively high NH loss rates. As such, the ratio bias is sensitive to the OH distribution used in the 3D model simulation.

The trend in the ratio for MCF is driven by the change in the spatial distribution of MCF after the emission drop in the mid-90s. Before the drop, the IH gradient of MCF was emission-driven and high (25%). This resulted in a negative correlation between OH/temperature and MCF in the NH, which drives the initially lower loss ratio. After the emission drop, the IH gradient becomes largely sink-dominated, and drops to 3%. The ratio then becomes similar, though not identical, to that of $CH_4$, which also has a relatively low IH gradient (5%). The exact reasons for the IH asymmetry in the OH loss rate are complex: further details are discussed in Supplement 3.

The derived IH OH ratio is sensitive to the demarcation of the two tropospheric boxes. If we shift the position from the equator to 8°N all IH OH ratios are reduced by 10 to 15%. However, the offset between the physical IH OH ratio and the actual loss ratio remains similar, as does the trend in the loss ratio for MCF.

### 3.1.4 Loss to the stratosphere

Figure 5 shows the stratospheric loss rate, as derived from Equations 1 and 7. Most notably, the stratospheric loss rate shows a significant negative trend for MCF. The MCF lifetime with respect to stratospheric loss in 1990 as calculated from TM5 is similar to the range reported in literature: 40 to 50 years (Naik et al. (2000); Ko et al. (2013)). Afterwards however, the corresponding timescale for stratospheric loss quickly increases. As loss to the stratosphere is a secondary loss process, it is generally assumed that variability in MCF loss is driven predominantly by OH variations (Montzka et al. (2011); Turner et al. (2017); Rigby et al. (2017)). Here, we find that this is not necessarily the case. The global lifetime of MCF with respect to stratospheric loss actually decreases by 5 to 10%, because of higher relative abundance of MCF in the stratosphere with respect to the troposphere. Thus, the increased tropospheric lifetime identified in Figure 5 is indicative of a slow-down in the troposphere-stratosphere exchange of MCF, rather than a reduced in-stratosphere loss. This decline in loss to the stratosphere is not an artefact resulting from treating a transport process as a loss process: if we take the exchange proportional to the



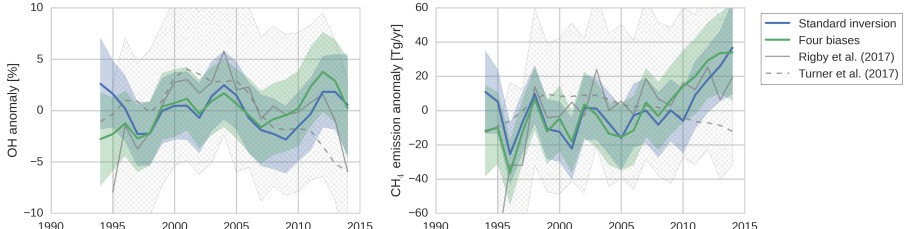

**Figure 6.** The results of two inversions of the two-box model: tropospheric OH anomalies (left) and $CH_4$ emission anomalies (right). In the standard inversion, we keep IH transport, NH/SH OH ratio and stratospheric loss of MCF constant, and we use NOAA observations. In the second inversion, we implement all four bias corrections instead (as described in Section 2.4). Both the mean anomalies and the 1-standard deviation envelopes are shown, where anomalies are taken relative to the time-averaged mean in each respective ensemble member. Plotted in grey are the anomalies as derived by Rigby et al. (2017) (from the NOAA dataset) and by Turner et al. (2017) (from a combined NOAA+AGAGE dataset), adjusted so that they, too, average to zero. The 1-standard deviation envelope from the Rigby et al. (2017) estimate is hatched in grey.

troposphere-stratosphere gradient we confirm the strong decrease in the exchange rate (results not shown). Previous research has identified that the tropospheric lifetime with respect to stratospheric loss could be decreasing (Krol and Lelieveld (2003); Prinn et al. (2005); Bousquet et al. (2005)), but not to the degree that we find here, and not relative to the troposphere-stratosphere gradient. This is important, because it means that also a three-box model with an explicit stratospheric box, such
as in Rigby et al. (2017), would not capture the decline.

The explanation we suggest for the increase in MCF lifetime with respect to stratospheric loss has to do with the nature of troposphere-stratosphere exchange, which consists of an upward and a downward flux. In practice, as MCF emissions decrease, the troposphere starts to transport air to the stratosphere which was exposed to lower MCF emissions, while the stratosphere is still transporting older air back to the troposphere (in the downward branch of the Brewer-Dobson circulation
(Butchart, 2014)) that was exposed to higher MCF emissions. Therefore, the delay between the two opposed fluxes results in a reduced net upward flux rate in an atmosphere with decreasing emissions compared to an atmosphere with increasing or constant emissions. Consistent with this hypothesis, we find that the stratospheric loss rate does not decrease when we fix MCF emissions at 1988 levels in a TM5 simulation, and that stratospheric loss does decrease, but recovers, when we fix emissions at 2005 levels over the entire analysis period (results not shown). This also implies that the troposphere-stratosphere exchange
will recover when MCF emissions stop decreasing.

For $CH_4$, we find a stratospheric lifetime of 160-170 years, similar to the range reported in Chipperfield and Liang (2013). For $SF_6$, there is no loss process implemented in our model. However, storage of $SF_6$ in the stratosphere acts as an effective loss process in the troposphere, with a lifetime of 100-160 years.





| Implemented bias(es) | MAE OH | OH trend | $\tau_{OH}$ MCF | $\tau_{trop}$ CH$_4$ | CH$_4$ emissions |
|---|---|---|---|---|---|
| | [%] | [%/yr] | [yr] | [yr] | [Tg/yr] |
| None/Standard | - | -0.04 $\pm$ 0.15 | 5.7 | 9.2 | 490 |
| Interhemispheric transport | 1.07 | 0.02 $\pm$ 0.14 | 5.9 | 9.4 | 478 |
| Surface sampling | 0.85 | 0.09 $\pm$ 0.15 | 6.0 | 9.6 | 469 |
| OH ratio | 0.68 | 0.04 $\pm$ 0.15 | 5.5 | 8.7 | 514 |
| MCF stratospheric loss | 0.68 | 0.04 $\pm$ 0.14 | 5.3 | 8.6 | 523 |
| All four | 1.28 | 0.19 $\pm$ 0.15 | 5.5 | 8.8 | 507 |

**Table 3.** Five metrics describing the outcome of the two-box inversions. The two-box inversions described are the standard set-up, four inversions with one bias implemented, and one inversion with all biases implemented. From left to right: 1) Mean absolute error (MAE) in OH anomalies between the standard inversion and each respective inversion. 2) Trend in OH over the 1994-2015 period 3) Mean lifetime of MCF with respect to OH (tropospheric burden MCF/loss to OH). 4) Mean total tropospheric lifetime of CH$_4$ (tropospheric burden CH$_4$/total loss CH$_4$). 5) Mean annual CH$_4$ emissions (including the soil sink).

## 3.2 Two-box inversion results

In this section, we present a comparison between the results of the standard inversion and an inversion that accounts for the four biases (referred to as "four-biases") . The inversion set-ups are described in Section 2.4. The OH and CH$_4$ emission anomalies of both inversions are presented in Figure 6, along with uncertainty envelopes of one standard deviation. The envelopes are wide,

and with respect to these envelopes there are no significant differences between our two inversions. Interestingly, differences between the two inversions are the smallest in the 1998-2007 period, during which MCF provides the strongest constraints on OH (Montzka et al., 2011). Note that the final analysis period starts from 1994 (rather than from 1990), because we only had sufficient NOAA coverage of MCF available from 1994 onwards.

Shown in grey in Figure 6 are the anomalies derived by Rigby et al. (2017) (from the NOAA dataset) and by Turner et al.

(2017). The four inversions show qualitatively similar time-dependicies, and differences generally fall within one standard deviation, and always within two standard deviations. Differences with Turner et al. (2017) are largest, most notably after 2010, which can be expected since they use a combined AGAGE+NOAA dataset, whereas we only use NOAA data. In Rigby et al. (2017) it was shown that the use of a different dataset can result in different OH anomalies, though these differences were insignificant with respect to their uncertainty envelopes. Also visibly is the uncertainty envelope of one standard deviation from

Rigby et al. (2017), which is notably larger than our envelopes. This is likely due to a combination of the higher observational uncertainties and the higher number of optimized parameters adopted in Rigby et al. (2017). Further discussion of differences with these two studies is provided in Section 4.





It is illustrative to further investigate how the identified biases impact the results. For this purpose, Table 3 presents five metrics for each of the two inversions, as well as for inversions where we implemented the bias corrections one-by-one (taking standard settings for the other parameters).

The first metric is the mean absolute error (MAE) in the OH anomalies between each respective inversion and the standard
inversion. The MAE provides an estimate of how much the OH estimate in a given year is affected by accounting for the bias. The highest MAE of 1.3% is small compared to the full envelope of each individual OH inversion (5-8%). This means that in terms of interannual variability over the entire period, the outcome is not much affected by the biases. However, as most biases show their strongest trends over short periods, the peak values of the differences between inversions even out somewhat when averaging over the entire period.

Secondly, we derived an OH trend for each inversion set-up. As described in Section 2.1, we mapped the uncertainty of each inversion set-up in a Monte Carlo ensemble of inversions. We fitted a linear trend to the derived OH timeseries of each ensemble member. From the resulting collection of linear fit coefficients, we derived a mean linear fit coefficient and its standard deviation. Differences between the OH trends derived from the different inversions are insignificant, similar as for the first metric. However, it is interesting to see that when all four biases are combined, we derive a shift to more positive OH
trends. In the standard inversion, 37% of the ensemble shows a positive trend, whereas in the four-bias inversion 92% of the ensemble shows a positive trend.

The final three metrics are the tropospheric lifetime of MCF with respect to OH ($(k_{MCF+OH}[OH])^{-1}$, as in Equation 1), the total tropospheric lifetime of CH$_4$ ($(k_{CH_4+OH}[OH]+l_{other})^{-1}$, as in Equation 1) and the derived global mean CH$_4$ emissions, averaged over the 1994-2015 period. Naturally, these three are strongly correlated. When we compare the relative differences
in for example the lifetime of MCF with respect to OH between different inversion set-ups to the MAE in anomalies, it is clear that the systematic offset between the different inversions (up to 10%) is much higher than the differences in anomalies (up to 1.3%). This is similar to what was seen for the biases themselves, where the systematic component tends to be much higher than the temporal variations (e.g. the bias in the IH OH ratio, shown in Figure 4). Especially significant is the case where a decrease in stratospheric loss of MCF is accounted for, which translates to higher OH and thus to shorter lifetimes of
CH$_4$, and finally to higher CH$_4$ emissions. The other biases mostly impact the IH gradient of MCF and of CH$_4$. For example, we have found that aggregating observations from the NOAA surface network to hemispheric averages results in a systematic overestimate of the true IH gradient (see Figure 3). To sustain this higher IH gradient the inversion will derive emissions that are also too high, and correcting for this bias (by reducing the IH gradient) results in lower CH$_4$ emissions: a difference of, in this case, 21 Tg/yr (surface sampling bias in Table 3).

**4   Discussion**

It is striking that the global CH$_4$ emissions we derive are significantly lower than those reported in literature. Firstly, in the two-box inversions of Turner et al. (2017) and Rigby et al. (2017), CH$_4$ emissions of 580-600 Tg/yr are found. The important difference with the model used by Turner et al. (2017) is that there the atmospheric mass is taken as the global atmospheric mass





$(5.15 \cdot 10^{18}$ kg), whereas we use the tropospheric mass $(4.4 \cdot 10^{18}$ kg). Thus we dilute all emissions in a 20% smaller air mass. When we run our two-box model with the global atmospheric mass, we find emissions of 600 Tg/yr in our standard set-up. In the model of Rigby et al. (2017), emissions are released in the tropospheric volume, as they explicitly include a stratospheric box containing 20% of the global atmospheric mass. However, if we adjust our a priori assumptions, we can also in large part

bridge the gap with their higher estimate of global $CH_4$ emissions. Specifically, when we adopt an IH exchange rate and an IH OH ratio similar to theirs (1.4 $yr^{-1}$ and 1.07 respectively) in our standard inversion, we find global $CH_4$ emissions of 563 Tg/yr, which is much more in line with their estimate.

Not only are our $CH_4$ emissions low compared to other two-box model studies, but they are also low compared to 3D modelling studies. This is important, because in our four-bias inversion we attempt to account for the most important two-box

model biases. Saunois et al. (2016) derived $CH_4$ emissions from 30 3D model inversions, and found emissions of 558 [540-570] Tg/yr over the 2000-2012 period: significantly higher than our best estimate of 507 Tg/yr (1994-2015). Bousquet et al. (2006) performed a full 3D inversion of $CH_4$, using OH fields that were optimized using MCF in a separate 3D model inversion (Bousquet et al., 2005). There, $CH_4$ emissions of $(525 \pm 8)$ Tg/yr were found over the 1984-2003 period, which is again higher than our best estimate, especially since the renewed $CH_4$ growth is not included. Part of the reason for the difference can be

found when studying the outcome of our TM5 simulations. In our standard 3D simulation, the IH gradient of MCF tends to be overestimated compared to observations from the NOAA network up to 2005, while global mean mixing ratios tend to be captured much better in this period. Following the line of reasoning we use in our two-box model, an inversion would result in lower MCF emissions and lower OH to reduce the IH gradient while not affecting the global growth rate. If we use lower OH in the 3D simulation, the same $CH_4$ growth rates will of course also require lower $CH_4$ emissions. Thus, the lower emissions

we derive in our two-box model are qualitatively consistent with what we see in our full 3D simulation.

There are several possible explanations for the difference with $CH_4$ emissions reported in literature. Firstly, the OH fields (from Spivakovsky et al. (2000), scaled with a factor 0.92 (Huijnen et al., 2010)) and MCF emissions (from the TransCom-$CH_4$ protocol, based on McCulloch and Midgley (2001)), that we use in our 3D simulation, could be too high. This would imply that our low two-box model estimate of global $CH_4$ emissions is more realistic than the higher emissions we use in our 3D

simulation. Secondly, a higher fraction of MCF emissions could be located in the SH ( 15-20% instead of  5-10%), in which case the MCF IH gradient will be reduced without having to lower emissions or OH. Thirdly, IH exchange in TM5 could be too slow, so that the IH exchange we use in our two-box model would also be too slow. This would explain the overestimate of the MCF IH gradient in our 3D simulation, without a need for adjusting emissions or OH. A combination of the last two points would also arise if MCF emissions moved from NH mid-latitudes to NH low-latitudes (e.g. India), since low-latitude

emissions will be exchanged more rapidly with the SH. At this point it is not clear which of these three explanations is most likely.

Similar to $CH_4$ emissions, there are large differences between absolute MCF emissions derived in different inversion set-ups. In our four-bias inversion, we find significantly lower MCF emissions ($\sim 10-30\%$) than the prior estimate based on emission inventories, with the exception of the 2010-2014 period. Liang et al. (2017) derived MCF emissions from the IH gradient, and

also found these to be systematically lower than those based on bottom-up industrial inventories. Possible explanations for



these differences are discussed above, but it does beg the question how useful a priori knowledge of MCF emissions is, when applied in a two-box model, as it seems difficult to correctly capture the absolute emission magnitude in this set-up.

As is acknowledged in the previous two-box inversion studies of OH (Rigby et al. (2017); Turner et al. (2017)), the problem of deriving OH from MCF and to a lesser degree from $CH_4$ is strongly underconstrained. Therefore, many solutions fit the problem almost equally well. Moreover, a best estimate, or most likely solution derived from a two-box model is a function of uncertain input parameters. For example, if a priori it is assumed that OH can only vary within a small band of 2%, then a most likely solution with small OH variations will be found. In this study, we have identified a number of parameters which show variations outside of conventionally assumed bounds. As such, for these parameters, the variations we find are never fully explored in a conventional two-box model inversion: even if done as comprehensively as in Rigby et al. (2017). A clear example is stratospheric loss of MCF, which is generally assumed to have only small variability (10 to 20%). Here, we find a persistent 70% drop in loss of MCF to the stratosphere. Similarly, we find persistent interannual variations in transport of MCF of up to 20%, compared to a conventional uncertainty of 10%. Thus, these are not scenarios that are readily explored within the conventional uncertainties of a two-box model inversion. In the end, these parameters do impact the outcome of the inversion. In the 1994-1998 period, during a period of strong redistribution of MCF, the impacts of the various biases are quite high, though when the four biases are combined in one inversion they partly cancel. During the 1998-2007 period, derived OH is less sensitive to the derived biases, likely due to a combination of a small role of uncertain emissions in the MCF budget (Montzka et al., 2011) and a period of relatively small redistribution of MCF. After this period, uncertainties increase, as the potential of emissions becomes important again. Constraints on emissions in the 2007-2014 period are mostly driven by the MCF IH gradient, which is impacted by the IH transport bias we have found. Interestingly, the period of relatively well-constrained OH (1998-2007) is not directly reflected in notably reduced uncertainty envelopes in either Rigby et al. (2017) or in our study. Finding the underlying reason is complex, as the final uncertainty envelopes result from non-linear interaction between the many parameters involved. However, it is another indication that interpretation of two-box inversion outcomes can be tricky.

Another crucial parameter in the two-box inversion is the uncertainty in the global mean mixing ratios, and in the IH gradient, as these uncertainties quantify the information content of the observational records. As we assume a fixed uncertainty of 10 Gg/yr in MCF emissions (as in Rigby et al. (2017)), constraints on OH from the MCF emission inventory reduce over time. As a consequence, the IH gradient of MCF provides a growing constraint on OH and thus the uncertainty in this gradient becomes a key parameter in the inversion. We provide an independent estimate of the uncertainty using 3D model output in Section 3.1.2, summarized in Table 2. These derived errors are somewhat sensitive to the set-up of the 3D simulation. For example, the uncertainties in the global growth rate and in the rate of change of the IH gradient, derived from the standard 3D simulation of both $CH_4$ and MCF (given in Table 2), become 20-30% higher in the nudged simulation, likely because the source-sink distributions in the nudged simulation are less regular. However, in order of magnitude our best estimate for this uncertainty remains unaltered. We can then compare the uncertainties we find to observational uncertainties as derived from bootstrapping by Turner et al. (2017). They find uncertainties in hemispheric means of 6-8 ppb for $CH_4$ and of 5-6% for MCF. Clearly, this is much higher than what we find and their uncertainties seem a gross overestimate considering the limited sensitivity of our result to a different source-sink distribution. In their most likely solution derived OH variations are such that the observed



post-2007 renewed growth of $CH_4$ coincides with a persistent decrease in $CH_4$ emissions. This solution does not fall within the uncertainty envelope we derive here (right panel in Figure 6). The difference in observational uncertainties is likely an important reason for this: their solution corresponds to a statistical inversion framework where information is weighted very differently.

In the end, conclusions from our study and those drawn by Rigby et al. (2017) and Turner et al. (2017) remain qualitatively similar. The post-2007 renewed growth of $CH_4$ need not be caused by a sudden increase in emissions in 2007. Rather, emissions could have increased more gradually over the 1994-2007 period, while $CH_4$ growth was suppressed temporarily by elevated OH levels. As seen in Figure 6, our inversion tends to indicate an increase in $CH_4$ emissions somewhere after 2007 more than Turner et al. (2017) and Rigby et al. (2017) do. With regards to Turner et al. (2017), this is likely because of different observational

uncertainties, while in Rigby et al. (2017) more variables are optimized (e.g. $k_{IH}$ and loss of MCF to the stratosphere), which also results in a more underdetermined inversion set-up. We instead prescribed these parameters, to focus on the impact of biases therein based on our full 3D simulations. Additionally, our MCF emission model can redistribute emissions relatively freely between years, which reduces interannual OH variations.

    In another recent study, an effort was made to find tracer alternatives to MCF (Liang et al., 2017). For this, their suggested

method was to use 3D model output to improve the results of a two-box model through intelligent parametrizations. Clearly, this is similar to the work described here. For example, similar to us, they find different IH exchange time scales for different tracers. However, our approach differs in that we explicitly resolve the two-box model in the 3D framework, while their study focuses mostly on fitting parameters empirically to find a match between two-box and 3D model results. In the end, we identify some issues not identified by Liang et al. (2017), and we find contradicting results on other issues. For the former, the trend in

statospheric lifetime for MCF and the different IH OH ratios of different tracers are not discussed by Liang et al. (2017). As for contradicting issues, the trend in IH transport ($CH_4$ and MCF) and in the surface sampling bias (MCF) are not found in their study. As we show here, these differences can be important. For example, in Liang et al. (2017) a strategy is described where two tracers are predominantly used to derive the IH OH ratio, which can then also be used for other tracers. Our work suggests that there should be careful consideration of different OH ratios seen by different tracers, and potential trends therein. A two-

box inversion is sensitive to the IH OH ratio, and we've shown that the IH OH ratio a tracer experiences depends strongly on that tracer's source-sink distribution.

    It is worth noting that the TM5 model, on which the two-box parametrization is based, has its own limitations, and so has treating TM5 as 'the truth'. For example, our simulations were done on the coarse horizontal resolution of $6° \times 4°$. This will impact how well NOAA background sites are actually situated in the background: for example, Mace Head is located in the

same grid box as the rest of Ireland. Similarly, any transport model is susceptible to some form of transport errors, and using a different 3D model for the two-box parametrization will likely result in different parameters. Therefore, we are careful in suggesting quantitative interpretation of our results. Certain aspects of the biases, such as a slow-down of MCF loss to the stratosphere and the strong variations in IH transport of MCF, are likely to also be found in other 3D transport models, as they are a direct consequence of the MCF emissions drop. Other aspects, such as the exact interannual variations of IH transport

of $CH_4$, or the 7% offset between the physical OH ratio and the effective OH ratio, should be interpreted with more care, as




these more strongly depend on the input emission and loss fields, and on the exact treatment of transport in the 3D model. However, the potential and the magnitude of the biases are real, and TM5 is a good starting point for exploring them, as TM5 has provided a strong basis for a great variety of studies in the past (e.g. Alexe et al. (2015); Laan-Luijkx et al. (2015); Bândă et al. (2016)).

## 5   Summary and Conclusions

In this study, we investigated variations in the global atmospheric oxidizing capacity, in conjunction with variations in the global $CH_4$ budget. We specifically revisit the use of two-box models to infer information about these quantities using global observations of MCF and $CH_4$.

We have shown how the transition from a 3D transport model to a two-box model affects a two-box model inversion. We identified a number of challenges in adopting a two-box model, some of which are known and obvious (IH transport; surface sampling bias), while others are less so (stratospheric loss; IH OH ratio). Two-box model parameters for these processes that were quantified from full 3D model output showed strong temporal trends mainly for MCF, which have not been identified in any previous research. In general, the biases result from a combination of variations in transport and variations in the spatio-temporal source/sink distributions of each tracer. We zoomed in on one such bias: the difference between the IH OH ratio seen by each respective tracer and the physical IH OH ratio in TM5 (see Supplement 3). This analysis showed that this bias is the net result of many compensating effects in the relative spatio-temporal distributions of all parameters involved (temperature, OH, MCF and $CH_4$). This is an example of how complicated biases resulting from the use of a two-box model can be. As a result, it is difficult to extrapolate our findings to other tracers without explicitly performing a similar 3D model analysis for these tracers.

When the impact of each of the biases is tested in a two-box model inversion, a number of things become clear. Firstly, as expected, we find that absolute OH and thus absolute $CH_4$ emissions show large deviations between the different inversions ($\sim 10\%$). Given that large parts of these deviations are constant through time, they do not necessarily impact conclusions of past two-box modelling studies that focused on interannual variations. However, it does beg the question how useful a priori knowledge on MCF emissions is, when applied in a two-box model, as it seems difficult to correctly capture the absolute emission magnitude in this set-up. Compared to the absolute differences, we find only small differences in OH anomalies (up to 1.3%, averaged over 1994-2015) relative to the full uncertainty envelope found here or by Rigby et al. (2017) (5-8%). Finally, we find that the conclusions one can draw from each individual inversion can be significantly affected: in the standard inversion only 37% of our Monte-Carlo ensemble showed a positive trend in OH over the 1994-2015 period, compared to 92% in the four-bias inversion. Thus, accounting for the biases increases the tendency to a positive trend in OH over the 1994-2015 period.

In the end, our work does not invalidate any of the conclusions of the two studies that served as incentive for this analysis (Rigby et al. (2017); Turner et al. (2017)), as the impact of the biases fall within known uncertainty bounds. Moreover, especially in the 1998-2007 period, MCF turns out the be a remarkably good tracer if one is interested in the impact of OH





variations on the $CH_4$ budget, since during this period biases for MCF and $CH_4$ behave similarly. On the other hand, the biases do contribute to the already significant uncertainty in derived OH, and properly accounting for them can be a piece in the puzzle of improving constraints on OH.

Moving forward, a likely next step is to incorporate more tracers in an effort to further tighten constraints on OH. In such a scenario, the tracer-dependent nature of the biases will likely increase the bias impact, and a proper 3D model analysis for each tracer becomes even more important. While efforts have been made to do so (Liang et al., 2017), the less-than-obvious nature of some of the biases we find makes such a 3D model analysis cumbersome. Therefore, we deem it important that a multi-tracer inversion in a full 3D model should also be performed, similar to the 3D inversion of MCF performed by Bousquet et al. (2005), but for more recent years. As an added advantage, a 3D model analysis would increase the pool of potential tracers that can be implemented to constrain OH. For example, the short-lived tracer $^{14}CO$ has been identified as a potential tracer to constrain OH (Quay et al. (2000); Krol et al. (2008)), but would not be implementable in a two-box model.

*Competing interests.* No competing interests are present.

*Acknowledgements.* This work was carried out on the Dutch national e-infrastructure with the support of SURF Cooperative. This work was funded through the Netherlands Organisation for Scientific Research (NWO), project number 824.15.002.



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
