# Peer review of "Constraints and biases in a tropospheric two-box model of OH"

_Atmospheric Chemistry and Physics, 2018_

## Referee Comment (RC1) · Anonymous Referee #2 · 14 Sep 2018

Review of Naus et al., 2018

This paper investigates the use of tropospheric two-box models for estimating global tropospheric OH concentrations and methane ($CH_4$) emissions. Output from a 3D atmospheric model (TM5) was used to investigate a range of potential biases brought about through the simplifying assumptions inherent in a box model framework. In particular, the influence simple parameterisations of inter-hemispheric transport, stratospheric loss and inter-hemispheric differences in OH are investigated, along with the ability of sampling networks to represent the hemispheric averages used in two-box models. This work follows from some recent publications that have used such models to propose that changes in OH, whilst being highly uncertain, may have contributed to some of the recent variability in $CH_4$. The findings of Naus et al broadly agree with the

inter annual changes inferred in these studies. However, this paper highlights the limits of such analysis and gives an indication of the impact of the above-mentioned biases on the outcome of such inversions.

The paper is topical and makes a valuable contribution to the recent literature on methane and OH variations and should be suitable for publication in ACP, following some changes.

General comments:

1. While the paper is generally well written, it is quite long and at times a little too verbose. This is partly because some of the issues are technical and, for the most part, are rightly discussed in depth. I've made a few suggestions where the text could be clarified or shortened below. However, I would encourage the author team to take another look over the manuscript as a whole and try to re-structure and re-focus parts of the text and shorten some bits. 2. Wouldn't the site-specific mole fractions from the 3D model be biased high because of the presence of "local pollution" in the model? As I understand it, NOAA conditionally sample data to select background mole fractions where possible, and then additional filtering is applied to remove obviously "polluted" samples. In contrast, the 3D model, particularly at the coarse resolution used (and I'm assuming for monthly mean output?), will almost always be influenced by nearby sources at most NOAA sites. The authors should at least mention this. Ideally, local pollution could be removed in the model mole fractions. 3. In Section 2.4, I think it might be worth clarifying how the "combined biases" run was done. I assume this was from a single model run, outputting all time-dependent quantities, rather than summing the individual biases? 4. Am I wrong to be left with the impression that, in order to avoid these biases, rather than throwing away 2-box models entirely, we could just re-parameterize them based on the outputs of 3D models (as was done in the inversions in this paper)? There are still good reasons why we might want to do this. For example, Rigby et al. (2017) used an MCMC approach which required many thousands of model evaluations. This is challenging with a 3D model, but fairly trivial with a box

model. Therefore, if we could use a small number of 3D model runs to derive better parameters, we could still use the advantages of a Monte Carlo inverse method (e.g. non-Gaussian distributions, non-linear models, etc).

Specific comments:

Title: "tropospheric two-box model of OH". Perhaps this should be OH, CH4, and methyl chloroform? Or just "in a tropospheric two-box model".

P1 L2 and several other places ("In two recent studies. . ."). A box model was also used in McNorton et al., 2016. However, I believe this was a 1-box model. Do you also want to discuss this? Or limit your discussion strictly to two-box models?

P1 L8: It's not quite accurate to say that two-box model studies use "fixed model parameters for interhemispheric transport, chemical loss rates and loss to the stratosphere". E.g. in Rigby et al., 2018, interhemispheric transport was allowed to vary each year (even though it didn't have any a priori interannual variability), and stratospheric loss depended on the stratospheric MCF or CH4 concentration, etc.

P1 L16 ("However, . . ."). This sentence is somewhat of a non sequitur. What is it in comparison to? E.g. are you saying that there is no overall trend without all biases?

P1 L17 ("Moreover, the magnitude. . . "). Perhaps say "absolute magnitude" to make this sentence more clear?

P1 L20 – L25: Given that the main results on OH and CH4 anomalies aren't too dissimilar from the two-box model studies compared to the other uncertainties (Figure 6), is it fair to say that it is "crucial" to use a 3D model? (Also see general comment 4).

P2 L3: The use of "resorts" makes it sound as if models are only used as a last resort. They are essential to understanding the data.

P2 L4: "most involving 3D transport model". Perhaps change to "more complex 3D transport models".
P2 L10: What does "explicitly contain much information on a species' distribution" mean? Does this mean that there is no information on longitudinal gradients in mole fractions, etc. If so, what is this trying to imply?

P2 L14: suggest change to ", which is largely determined by the TROPOSPHERIC hydroxyl radical CONCENTRATION"

P2 L18: "consequences this has had in the past" (grammar). However, perhaps this sentence can be cut as it doesn't really say anything.

P2 L22: suggest "more robust observational constraints on OH on the large scales are THOUGHT TO BE derived indirectly…" because we don't really know this for sure.

P2 L25 (e.g. Montzka et al., 2000; Bousquet et al (2005)). I suggest referencing some of the earlier papers here (e.g. Lovelock, Prinn)

P2 L30: Not sure why Montzka et al is described as the "most important" work here? Seems unnecessary. Also, note that the results of Montzka et al are consistent with Rigby et al., 2017.

P3 L10: "do not significantly affect the outcome". I don't think you know this for sure, so suggest changing to "are not thought to significantly affect the outcome"

P3 L12: Perhaps change "approximating interhemispheric (IH) exchange using SF6" to "approximating the possible range of interhemispheric exchange rates using SF6" or similar, because Rigby et al. only used SF6 to estimate the a priori value of IH exchange rate (although, not surprisingly, these values weren't strongly updated in their inversion).

P3 L7 – L30: I believe these paragraphs are trying to state the reasons why you'd want AT LEAST a two-box model (i.e. so that you can examine the trend and IH gradient). Is that right? Otherwise, it's not clear why these two criteria are described in detail, rather than other factors (e.g. computational efficiency)

P3 34: "reversely" suggest change to "conversely"

P4 L5 – L20: This paragraph is a little hard to follow as it concerns two different parts of the study, the first of which is subdivided into four sub-sections. For the latter, I suggest using "firstly", "secondly", "thirdly", "fourthly" to make it clear where this section ends and the next begins. Or restructure in some other way.

P5 L3: HˆT. The T should not be bold.

P6 L2: suggest change to "leave more freedom with respect to the timing of emissions". However, I'm not sure what this assertion is based on? Is this justified? If so, how?

P7 Table captions: suggest change "perturbed in Monte Carlo" to "perturbed in Monte Carlo ensemble" or something more descriptive.

P8 L18: The nudged simulation needs to be described in a little more detail here (but still succinctly). It's not obvious what distinguishes it from the other analysis from the description in this paragraph.

P8 L29: Perhaps not obvious what is meant by "budget" in this context. The sum of the sources minus the sinks in each box? If so, perhaps just say so.

P9 L2: "Note that we have do not" (delete "have"?)

P9 L31: "that deviated significantly from what is generally expected". This is a very vague sentence. I'm not sure what the authors are trying to say here. Are they saying that this is based on their intuition?

P10 L10: "has undergone" instead of "undergoes", which suggests that MCF is redistributed repeatedly?

P10 L18: ". . . surface measurements do not inform much on vertical gradients". Not entirely sure what this means (seems obvious), but re-wording is at least required. Perhaps delete.

P10 L20: "but this information is difficult to correctly incorporate. . ." Why would this be the case?

P10 L32: suggest "We explore this bias. . ."

P11 Figure 1 caption: "but adjusted by correction factors". I suggest being more specific. What exactly has been done.

P11 L5: I'm not sure that it's fair to say that Rigby et al assumed a "constant stratospheric lifetime". The local lifetime in their stratospheric box was time invariant. However, this would mean that the transient lifetime varied as the distribution of MCF changed. Perhaps say that the local stratospheric loss rate was constant (could also say that the value of this loss was consistent with a steady-state lifetime of X years)?

P11 L13: "As mentioned in . . . would obscure the impact of the bias correction." I'm not sure what this sentence means. Needs to be more specific.

P12 L4: "Equations 1 and 7". Perhaps it's better to reference Section 2.2.2, which actually describes how this calculation was done?

P12 L8: perhaps "SF6 has no SIGNIFICANT sink" or similar (not significant over these timescales, but it does have a sink)

P12 L13: "changes sign (NOT SIGNS) in the tropics". I'm not sure what is meant by this. Do you mean that the gradient changes sign seasonally now? If so, why just in the tropics?

P12 L18: Does SF6 have annually repeating sources in the model runs too? Would be good to remind us here.

P12 L20: What does "the treatment of this data in TM5" mean?

P13 L21 – L26: This seems relatively important. Worth showing as dotted lines in Figure 2?

P13 L28: Reference Figure 1 as well as Figure 3?

P13 L30: should this be "tropospheric hemisphere" instead of "troposphere"?

P14 L8: "The shift in the bias for MCF is driven by the latitudinal dimension ". What does this mean?

P14 L8: "combined with simple latitudinal interpolation." Not sure what this interpolation refers to. EDIT: After reading the Supplement, I think I understand a bit better. However, it still needs briefly explaining in the main text.

P15 L9 – P16 L8: Is this long discussion of (random?) uncertainties really necessary? To me it seems that this is a side issue that distracts from the main message (the biases). Perhaps cut this, or shorten substantially.

P16 Figure 4: I suggest using a more descriptive y-axis label.

P16 L7: Should this be Rigby et al.. 2017, rather than 2013?

P18 L1: This is an interesting discussion. However, I'm interested in the relative magnitudes. Would treating the stratospheric loss as a function of the strat-trop gradient get us most of the way to removing the "bias" (or would the addition of the stratospheric box be a pretty good way forward)? Or do you really need resolved stratospheric transport in detail to address most of this issue?

P18 L15: I assume this change would happen over a period of years? The sentences makes it sound rather instantaneous.

P19 L14: "visible" rather than "visibly"

P22 L11: Is the use of "persistent" correct here? Aren't you referring to a transient effect. Also, similarly to comment on P18 above, do we know how much of this drop might be expected with a model that attempts to resolve the stratosphere (e.g. Rigby et al., 2017).

[Figure]

P22 L22: I'm not entirely sure what "persistent interannual variations in transport of MCF of up to 20%" means. Can you clarify?

P22 L22: I don't like the use of the word "tricky". Furthermore, couldn't the reasons for this due to more than just the use of box models? E.g. changes in magnitude and timing of emissions reduction?

P22 L25: Rigby et al. didn't assume a fixed emissions uncertainty of 10Gg/yr. They used an emissions model that had a lot of unknown parameters (including the possibility up to 10Gg/yr of unreported production).

P22 L34: Perhaps "gross overestimate" is too strong!

P23 L3 – 4: "where information is weighted very differently". This could do with clarification as I'm not entirely sure what is meant by this.

P24 L2: "However, the potential and the magnitude of the biases are real". Please clarify.

Supplement:

S1: Wouldn't it be safer to omit months where there was a significant fraction of missing data, rather than interpolate from nearby stations?

S3.5: Perhaps choose and different subheading that "conclusions" here. Make it more specific since it is not an overall conclusion.

References

McNorton, J., Chipperfield, M. P., Gloor, M., Wilson, C., Feng, W., Hayman, G. D., Rigby, M., Krummel, P. B., O'Doherty, S., Prinn, R. G., Weiss, R. F., Young, D., Dlugokencky, E. and Montzka, S. A.: Role of OH variability in the stalling of the global atmospheric CH4 growth rate from 1999 to 2006, Atmospheric Chemistry and Physics, 16(12), 7943–7956, doi:10.5194/acp-16-7943-2016, 2016.

---

## Referee Comment (RC2) · Anonymous Referee #1 · 22 Oct 2018

Naus et al. presented a modeling analysis to quantify the uncertainty in the derived estimates of global OH and CH4 emissions due to the difficulty in an accurate representation of the real atmospheric 3-Dimensional transport and chemical processes in a simplified two-box model framework. A comprehensive set of experiments were conducted to investigate the impact of inter-hemispheric transport, representative of surface observational network, inter-hemispheric OH ratio, the differences in the sensitivity of various chemical compounds to the spatial distribution of emissions and OH, etc. The analysis presented in this work is a further step of what have been discussed in the recent literature on global OH abundance and CH4 emission estimates, e.g. Rigby et al. (2017), Turner et al. (2017), Liang et al. (2017). Results from this work are a nice addition to these previous published papers and should be published after the

following comments, mostly minor, are addressed.

Major comments:

1. Use of tense in "Abstract" and "Summaries and Conclusions" are not consistent. I have noticed that the authors swap back and forth between present tense and past tense in these two sections. The common practice would be use consistently one tense through abstract and conclusions sections. For example, we did . . .; we found . . ., we investigated . . .

2. Personally, I found the current version of Abstract not fully capturing the essence of the findings from this work. While the results presented in the main body of the text are important additions of the existing literature, the abstract only includes rather general and vague discussions and no clear identifications of what are the crucial parameters that needed to be considered if one is to adopt the two-box model approach, etc. A reorganization of the abstract, with clear emphasis on what are the key findings of this, in the context of previous literature, would be helpful to readers.

3. P12, last paragraph and Figure 2. The positive trends in the IH exchange rate for CH4 and SF6 are very puzzling. Based on the results presented, it is not convincing to arrive at the conclusion that these trends are due to acceleration of IH transport of air mass or a shift in the pattern of IH transport. Are you sure this isn't due to a model spin-up related issue? If it is indeed the change in the IH transport rate, there must be a way to quantify this using the proper diagnostics from the TM5 model or set up sensitivity runs to tackle the problem. At this point, it sounds very hand-waving to arrive at the conclusion that it is due to changes in IH transport with no real analysis backing up the conclusion.

4. P15, 1st paragraph. As discussed in section 3.3.1 last paragraph in Liang et al. (2017), if one were to use the surface measurements to perform two-box model cal-culation, the NH surface to SH surface IH transport time is needed in the gradient-to-emissions calculation to be consistent. On the hand, if the entire tropospheric air mass

is considered, the IH transport timescale is significantly reduced (as demonstrated in this paper) due to the nature of cross-hemispheric transport in the troposphere. More details can be found in Liang et al, section 3.3.1. The authors seem to have the two methods (or concepts) mixed when discussing this. It would be good to add a discussion on these two different conceptual models if the authors wish to compare the results presented in this work with those discussed in Liang et al. to avoid confusion.

5. P22, L26-27. "In the end, conclusions from our study and those drawn by Rigby et al. (2017) and Turner et al. (2017) remain qualitatively similar". Isn't this a much more important conclusion than the way it is presented here in this paper? Despite all the other details discussed, e.g. box-model simplifications, bias corrections, etc., this paper confirms the findings from Rigby et al. (2017) and Turner et al. (2017). In addition, I found Sections 4 and 5 somewhat wordy. While lots of details are discussed, it is hard to drawn the main conclusions, e.g. what are the important details that one needs to consider when conducting box-model-based calculations. Some reorganization of the discussion and conclusions and emphasis on the key factors/uncertainties/parametrizations can be helpful.

Minor comments:

1. P1, L3: OH is already defined in the previous line.

2. P2, L 4: What do you mean by "involving"? It might be better to use "complex" or "state-of-the art"

3. P2, L30: has -> had

4. P3, L18: Add "Observation-inferred" before "emissions"

5. P3, L23-34. "these two processes have a very different effect on the IH gradient of MCF". It is much more helpful to stated directly what are the different effects of these two processes, than leave it to the readers to wonder about it.

6. In some places, inter-hemispheric is used while in others, interhemispheric (intrahemispheric) is used. Please be consistent.

7. P8, L11 and L13: subscript 4 in CH4

8. P9, L2: delete have after we

9. P9, L5: the 3D and the two-box models. (plural for model)

10. P9, L10: Add Global Monitoring Division (GMD) after NOAA.

11. P9, L30: "we identified three parameters . . .". Please state which three.

12. P15, L24. Add "it" before "is".

13. P22, L18-19. This sentence is awkward. Need rephrase.

14. P22, L19. Change to "uncertainties we found"

15. P23, L20. Change to "They found"

16. P22, L22. Add "," after "likely solution"

---

## Author Comment (AC1) · 21 Nov 2018

**Reply to reviewer #1**

We thank the reviewer for the  helpful comments, and are confident that the revisions encouraged by the reviewer have resulted in an improved manuscript. Below we address the main points put forward by the reviewer. We found no issue with the minor comments, and have implemented all of them in the revised manuscript.

1. *Use of tense in "Abstract" and "Summaries and Conclusions" are not consistent. I have noticed that the authors swap back and forth between present tense and past tense in these two sections. The common practice would be use consistently one tense through abstract and conclusions sections.*

We have corrected this.

2. *Personally, I found the current version of Abstract not fully capturing the essence of the findings from this work. While the results presented in the main body of the text are important additions of the existing literature, the abstract only includes rather general and vague discussions and no clear identifications of what are the crucial parameters that needed to be considered if one is to adopt the two-box model approach, etc. A reorganization of the abstract, with clear emphasis on what are the key findings of this, in the context of previous literature, would be helpful to readers.*

We have restructured the abstract, to better reflect the key findings of the paper.

3. *P12, last paragraph and Figure 2. The positive trends in the IH exchange rate for CH4 and SF6 are very puzzling. Based on the results presented, it is not convincing to arrive at the conclusion that these trends are due to acceleration of IH transport of air mass or a shift in the pattern of IH transport. Are you sure this isn't due to a model spin-up related issue? If it is indeed the change in the IH transport rate, there must be a way to quantify this using the proper diagnostics from the TM5 model or set up sensitivity runs to tackle the problem. At this point, it sounds very hand-waving to arrive at the conclusion that it is due to changes in IH transport with no real analysis backing up the conclusion.*

As mentioned in the manuscript, we ran a simulation with annually repeating meteorology. As expected, for $CH_4$, the IH exchange coefficient we derived from this simulation was near-constant, with no spin-up effects evident. Moreover, a simulation in which the meteorological fields of 1992 were repeated resulted in slower exchange than a simulation with 2012 fields. The only difference between those simulations and the simulations that give a trend in IH exchange are the meteorological fields that were used. This provides a solid basis for our conclusion that the meteorological fields (+ their treatment in TM5) are driving the trend. We discuss this issue now in more detail in Supplement 4, along with other sensitivities of the exchange rate.

4. *P15, 1st paragraph. As discussed in section 3.3.1 last paragraph in Liang et al. (2017), if one were to use the surface measurements to perform two-box model calculation, the NH surface to SH surface IH transport time is needed in the gradient-to-emissions calculation to be consistent. On the hand, if the entire tropospheric air mass is considered, the IH transport timescale is significantly reduced (as demonstrated in this paper) due to the nature of cross-hemispheric transport in the troposphere. More details can be found in Liang et al, section 3.3.1. The authors seem to have the two methods (or concepts) mixed when discussing this. It would be good to add a discussion on*

*these two different conceptual models if the authors wish to compare the results presented in this work with those discussed in Liang et al. to avoid confusion.*

The difference in approach is whether the exchange time is defined with respect to the surface gradient (as in Liang et al., 2017) or with respect to the tropospheric gradient (as in this study). As noted in Liang et al. (2017), our approach will likely result in faster IH exchange, and it is conceivable that there will also be time-dependent differences between the two approaches.

We cannot define our parametrization with respect to the surface gradient, because the 3D model budget would not be closed. Therefore, we cannot explicitly quantify how our results would be different if exchange is taken proportional to the surface gradient. However, it is possible that the large variations we find in the exchange rate of MCF could also be there for an exchange rate defined with respect to the surface gradient. Therefore, we deem it important to point out the difference in results, even if the two methods are not identical. If 3D transport models can provide us with inter-annual variability in IH exchange, then we should try to use this information.

To explain this issue more clearly, we added:

*Additionally, for the parametrization, Liang et al. (2017) used hemispheric mean mixing ratios derived from the surface network, whereas we based mixing ratios on the full hemispheric troposphere in TM5.* **P22 Lines 15-17**
*Some of the differences may be explained by the definition of hemispheric mean mixing ratio (surface-based versus full troposphere), but further reconciliation of the two approaches in future research is necessary.* **P22 Lines 22-24**

5. *P22, L26-27. "In the end, conclusions from our study and those drawn by Rigby et al. (2017) and Turner et al. (2017) remain qualitatively similar". Isn't this a much more important conclusion than the way it is presented here in this paper? Despite all the other details discussed, e.g. box-model simplifications, bias corrections, etc., this paper confirms the findings from Rigby et al. (2017) and Turner et al. (2017).*

We indeed confirm the results of previous studies, and we added a line to the Conclusions to acknowledge this more fully (see below). However, this could also be expected: due to the undetermined nature of the problem, improvements in any one parameter will not significantly improve constraints on OH. Despite this insensitivity, improving constraints on parameters one-by-one seems like an obvious way forward. Therefore, one of our conclusions is that we can use information from a 3D model to improve constraints on four parameters in the inversion. Even if this does not directly translate in improved constraints in the end-products (emissions, OH), it can be a useful first step. We think that for future research this is an equally important result, compared to agreement between two/three very underdetermined two-box model inversions.

*This indicates that significant uncertainties in parameters unrelated to the identified biases remain, and these uncertainties attenuate the impact of the biases on an inversion. As such, we confirm in large part the conclusions drawn by Rigby et al. (2017) and Turner et al. (2017) regarding the underdetermined state of the problem.* **P23 Lines 21-23**

*In addition, I found Sections 4 and 5 somewhat wordy. While lots of details are discussed, it is hard to drawn the main conclusions, e.g. what are the important details that one needs to consider when conducting box-*

*model-based calculations. Some reorganization of the discussion and conclusions and emphasis on the key factors/uncertainties/parametrizations can be helpful.*

Reviewer #2 commented similarly on the length of some parts of the manuscripts. Per your suggestions, we have restructured parts of the discussion and conclusion and sharpened our phrasing.

---

## Author Comment (AC2) · 21 Nov 2018

**Reply to reviewer #2**

We thank the reviewer for the helpful comments, and are confident that the revisions encouraged by the reviewer have resulted in an improved manuscript. Below we address first in detail the general comments. We also discuss the specific comments, in as far as our implementation and choices required further explanation. Points not mentioned were implemented without issue.

**General comments**

1. *While the paper is generally well written, it is quite long and at times a little too verbose. This is partly because some of the issues are technical and, for the most part, are rightly discussed in depth. I've made a few suggestions where the text could be clarified or shortened below. However, I would encourage the author team to take another look over the manuscript as a whole and try to re-structure and re-focus parts of the text and shorten some bits.*

Reviewer #1 similarly commented on the wordy nature of parts of the manuscripts. We have sharpened and shortened the manuscript, partly based on the reviewers' comments and partly based on our own insights. These adjustments mainly concerned the abstract, the discussion and the conclusions.

2. *Wouldn't the site-specific mole fractions from the 3D model be biased high because of the presence of "local pollution" in the model? As I understand it, NOAA conditionally sample data to select background mole fractions where possible, and then additional filtering is applied to remove obviously "polluted" samples. In contrast, the 3D model, particularly at the coarse resolution used (and I'm assuming for monthly mean output?), will almost always be influenced by nearby sources at most NOAA sites. The authors should at least mention this. Ideally, local pollution could be removed in the model mole fractions.*

For our analysis, TM5 was not sampled from monthly mean fields (!), but samples were taken by interpolation between the 3-hourly fields in time, and linearly in space. This means that as long as meteorology was simulated correctly, our model-sampled observations should be of air from clean air sectors, similar to the NOAA sampling strategy.

However, it's true that our TM5 simulations were done at the coarse resolution of 6x4 degrees, so that some additional sample pollution could occur due to colocation of sources with sample sites in the same grid box (though in principal spatial interpolation is implemented to correct for this).

For this reason, we performed a quality check of the model-sampled observations. For the check, we removed the trend and seasonal cycle from the $CH_4$ and MCF observational records, and we investigated the spread in the residuals per site as a proxy for pollution. We quantified this in the annual mean residual standard deviation (RSD), which will be higher in the presence of frequent pollution events. For both MCF and $CH_4$, we find that the RSD in the model-sampled and real-world records agree well at most sites. At a few sites the RSD of model-sampled observations is slightly higher than that of real-world observations (indicating over-polluted sampling in the model), but at others it is the other way around. As such, we did not find any evidence of systematically higher pollution of model-sampled observations, relative to real-world observations.

As this is an important issue, we now added a brief comment on the quality check to the manuscript :

*We checked that the TM5-derived observational timeseries were not systematically more polluted than the real-world NOAA-GMD observations. For this we detrended and deseasonalized the $CH_4$ and MCF timeseries per surface site, and quantified the spread in the residuals. At most sites, we found no offset between residual spread found in the TM5-derived versus the real-world timeseries. At a small number of sites, TM5-derived timeseries showed more spread in residuals, while at others the spread was less. Therefore, we find no evidence for systematic biases in TM5-sampled observations..* **P22 Lines 27-32**

3. *In Section 2.4, I think it might be worth clarifying how the "combined biases" run was done. I assume this was from a single model run, outputting all time-dependent quantities, rather than summing the individual biases?*

This is explained in Section 2.4. Indeed, to obtain the results for combined biases, we ran the two-box inversion with all four biases corrected for simultaneously. We feel we already cover this subject by treating it in its separate section. For example:

*Through comparison of the outcome of the standard inversion and an inversion with one or more biases implemented simultaneously, we can evaluate the individual and cumulative impact of the biases on derived OH and $CH_4$ emissions.* **P11 Line 20**

*For this purpose, Table 3 presents five metrics for each of the two inversions, as well as for inversions where we implemented the bias corrections one-by-one (taking standard settings for the other parameters).* **P19 Lines 1-3**

4. *Am I wrong to be left with the impression that, in order to avoid these biases, rather than throwing away 2-box models entirely, we could just reparameterize them based on the outputs of 3D models (as was done in the inversions in this paper)? There are still good reasons why we might want to do this. For example, Rigby et al. (2017) used an MCMC approach which required many thousands of model evaluations. This is challenging with a 3D model, but fairly trivial with a box model. Therefore, if we could use a small number of 3D model runs to derive better parameters, we could still use the advantages of a Monte Carlo inverse method (e.g. non-Gaussian distributions, non-linear models, etc).*

This is a fair point that was not sufficiently acknowledged in the original manuscript. Indeed, a two-box inversion allows incorporation of results from multiple 3D transport models, and this is an important advantage it holds over any one 3D transport model. Moreover, computational efficiency is a great advantage for many reasons. Therefore, we have added some additional discussion to emphasize this potential use of our analysis:

*The identified two-box model biases contribute to the already significant uncertainty in derived OH, and properly accounting for them can be a piece in the puzzle of improving constraints on OH. Moving forward, a likely next step is to incorporate more tracers in an effort to further tighten constraints on OH. In such a scenario, the tracer-dependent nature of the biases will likely increase the bias impact, and a proper 3D model analysis for each tracer becomes even more important. Already, efforts have been made to do so (Liang et al., 2017), and in this study we provide further suggestions for such an approach. A distinct advantage in this approach is that information from multiple 3D transport models can be used to tune the two-box inversion, making the inversion outcome less reliant on transport parametrizations of any single 3D transport model. Additionally, computational efficiency of simple models allows for complex statistical*

*inversion frameworks, incorporating, for example, hierarchical uncertainties (Rigby et al., 2017).* **P23 Lines 27-34**

However, as a counter-point there are also significant downsides to such an approach. For example, the inversion becomes sensitive to the settings used in the 3D transport model, e.g. source-sink fields. Sensitivity analyses with different OH fields and different source fields should be made in all the different model configurations. Additionally, if a two-box inversion suggests an adjustment to certain state parameters (e.g. emissions or OH), then it should be tested again whether the 3D model-derived bias corrections are sensitive to these adjustments.

Then there are issues that are difficult to resolve at all in a two-box model. For example, the latitudinal gradient of MCF minimizes in the tropics, post-2000. Thus, IH exchange of MCF is mostly driven by the slight IH asymmetry in this minimum, rather than by the overall IH gradient, which is the parameter that is optimized in a two-box model. Very likely for this reason, we find that the derived IH transport rate for MCF is sensitive to the demarcation of the two hemispheres. This uncertainty does not reflect any real uncertainty in the 3D transport model, but is rather an artefact resulting from the two-box parametrization. We find it hard to resolve this issue.

So while our analysis is indeed an example of how an approach suggested by the reviewer could be implemented, there are many challenges that remain before such an analysis actually resolves all of the problems we have identified. We encourage such work, of course, but in some ways a full 3D model inversion might be easier and more complete, and better represent reality.

**Specific comments**

*P1 L2 and several other places ("In two recent studies: : :"). A box model was also used in McNorton et al., 2016. However, I believe this was a 1-box model. Do you also want to discuss this? Or limit your discussion strictly to two-box models?*

We have added more emphasis in the introduction to the fact that we really mean to focus on two-box models. While there might be some overlap, most issues we've identified are specific to the two-box model and would be different in a one-box. Due to the usefulness of the IH gradient we really do think that future work regarding the problem of OH will also involve models of (at least) two boxes. We do now refer to McNorton et al. (2016) in more general terms, as the work is relevant to our study.

*P1 L20 – L25: Given that the main results on OH and CH4 anomalies aren't too dissimilar from the two-box model studies compared to the other uncertainties (Figure 6), is it fair to say that it is "crucial" to use a 3D model? (Also see general comment 4).*

The small impact on the final conclusions reflects the many uncertainties of the problem. Even if we know the four parameters we've derived exactly, as is assumed in our two-box inversion, we still find a very uncertain solution. In large part due to this large uncertainty of the final solution we find agreement with existing literature.

However, moving forward, the objective of future research will be to reduce the uncertainties on derived OH. Given that the final uncertainty results from uncertainty in many parameters, reducing the uncertainty on any one parameter will not solve the problem (as is reflected by our results). However, if incorporation of information from a 3D transport model allows us to reduce uncertainty in a few of these parameters, then that does seem like a crucial first step, even if the immediate impact is not directly noticeable.

This is also related to the reviewer's later comment on observational uncertainties. Again, even with lower observational uncertainties, the problem might still be strongly underdetermined. However, piece-by-piece the combination of these kinds of improvements should put us on the right track to converging constraints on OH. The fact that the large bias corrections and the large differences in observational uncertainties do not significantly affect the final solution, is a testament to how hugely uncertain the problem was to begin with.

*P2 L10: What does "explicitly contain much information on a species' distribution" mean? Does this mean that there is no information on longitudinal gradients in mole fractions, etc. If so, what is this trying to imply?*

We mean that there is little spatial information included in a one- or two- box model, e.g. the tropical maximum of OH is not captured. Many of the biases we derive are driven by non-linearities between a species' distribution and varying source-sink fields (see e.g. Supplement 3). Use of "explicitly" acknowledges that it might be included implicitly through parametrizations derived from a 3D transport model.

*P2 L25 (e.g. Montzka et al., 2000; Bousquet et al (2005)). I suggest referencing some of the earlier papers here (e.g. Lovelock, Prinn)*

We already referred to these studies in the next sentence.

*P6 L2: suggest change to "leave more freedom with respect to the timing of emissions". However, I'm not sure what this assertion is based on? Is this justified? If so, how?*

In Rigby et al. (2017), the emission model allows the emissions to be shifted between decades. In our emission model, the shifting occurs between years. Though there are different constraints on the shifting in our model, in practice our model still results in more freedom with respect to the timing of emissions.

Our emission model results in uncertainties that roughly agree with those reported in McCullogh and Midgley (2001). These uncertainties are very high in the years where production was phased out (e.g. a 2-sigma range of 15.0 to 65.1 Gg/yr in 1997), and so we sought to reflect these uncertainties in our model. Emission timing, to us, seems a very uncertain uncertainty. In the absence of conclusive evidence that the Rigby et al. (2017) emission model is a better reflection of the actual uncertainties, we will retain our current approach.

*P12 L20: What does "the treatment of this data in TM5" mean?*

This phrase relates to how meteorological fields (wind, temperature …) result in transport of tracer mass in TM5. An example would be the parametrization of convection, which can have a large influence on interhemispheric exchange (e.g. Tsuruta et al., 2016), and the pre-processing of the meteorological data to create mass-conserving transport in TM5 (Bregman et al., 2003). The point is that there has to be a trend in some meteorological parameters for the final tracer transport in TM5 to exhibit a trend, but that not necessarily every 3D transport model will show a similar trend in the end-product, due to different sensitivities to meteorological parameters. We deem this issue sufficiently covered in the manuscript.

*P13 L21 – L26: This seems relatively important. Worth showing as dotted lines in Figure 2?*

It is important. Also resulting from a comment by Reviewer #1, it seemed useful to visualize the effect of various sensitivity tests on the IH transport rate (hemispheric demarcation; nudging; annually repeating

meteorological fields). For this purpose, we have included the information in additional supplemental figures, so as not to overcrowd the figure in the main text (Supplement 4).

*P15 L9 – P16 L8: Is this long discussion of (random?) uncertainties really necessary? To me it seems that this is a side issue that distracts from the main message (the biases). Perhaps cut this, or shorten substantially.*

We think this is actually a very important issue. The gap between observational uncertainties used in Rigby et al. (2017) and Turner et al. (2017), and that reported by NOAA (at least for CH$_4$) is a factor 7. Given that the shared conclusion of the two studies is that the problem of constraining OH is strongly underdetermined, it seems crucial to resolve which observational uncertainty is correct.

From the TM5 simulations, we find much lower uncertainties than Rigby et al. (2017) and Turner et al. (2017), more in line with the observational uncertainty reported by NOAA, and also derived by ourselves. These lower observational uncertainties can be an important step in reducing the final uncertainty on derived OH variations (see also comment above).

*P18 L1: This is an interesting discussion. However, I'm interested in the relative magnitudes. Would treating the stratospheric loss as a function of the strat-trop gradient get us most of the way to removing the "bias" (or would the addition of the stratospheric box be a pretty good way forward)? Or do you really need resolved stratospheric transport in detail to address most of this issue?*

*P22 L11: Is the use of "persistent" correct here? Aren't you referring to a transient effect. Also, similarly to comment on P18 above, do we know how much of this drop might be expected with a model that attempts to resolve the stratosphere (e.g. Rigby et al., 2017).*

If we take stratospheric loss proportional to the tropospheric abundance, we find a reduction of 68% in the loss rate. If we take it proportional to the stratosphere-troposphere gradient, we find a reduction of 63%. We have added these numbers to the main text. Thus, the slow-down is really related to transport, and can hardly be corrected for by using a stratospheric box.

As for the use of persistent, we intend to say that it's multi-annual and not random year-to-year variability. Indeed, if at any point the emissions stop decreasing, then the loss rate will recover, so that it is indeed a potentially transient effect. We have clarified this in the text.

*S1: Wouldn't it be safer to omit months where there was a significant fraction of missing data, rather than interpolate from nearby stations?*

Omitting months from individual stations results in a changing surface network. This is very undesirable, as it results in large jumps in mixing ratios, which reflect the large systematic uncertainties in the global mean mixing ratios we derive. However, as long as these uncertainties are systematic, they have no impact on the derived growth rates, which provide the main constraint in our inversions.

While interpolation has its own uncertainties, it does circumvent the more significant uncertainties that a changing station network would cause. In general, we found that the site-to-site ratios we used are quite constant through time, as all site pairs are background sites located relatively close to each other (e.g. SPO/CGO and BRW/ALT). Finally, the (small) uncertainties we derived from the TM5 simulation show that the technique works remarkably well.

**References**

Bregman, B., Segers, A., Krol, M., E Meijer & van Velthoven, P. On the use of mass-conserving wind fields in chemistry-transport models. *Atmos Chem Phys 3, 447–457, 2003.*

McCulloch, A. and Midgley, P. M.: The history of methyl chloroform emissions: 1951–2000, *Atmospheric Environment, 35, 5311–5319, 2001.*

McNorton, J., Chipperfield, M. P., Gloor, M., Wilson, C., Feng, W., Hayman, G. D., Rigby, M., Krummel, P. B., O'Doherty, S., Prinn, R. G., et al.: Role of OH variability in the stalling of the global atmospheric CH4 growth rate from 1999 to 2006, *Atmospheric Chemistry and Physics, 16, 7943–7956, 2016.*

Rigby, M., Montzka, S. A., Prinn, R. G., White, J. W. C., Young, D., O'Doherty, S., Lunt, M. F., Ganesan, A. L., Manning, A. J., Simmonds, P. G., et al.: Role of atmospheric oxidation in recent methane growth, *Proceedings of the National Academy of Sciences, 114, 5373–5377, 2017.*

Tsuruta, A., Aalto, T., Backman, L., Hakkarainen, J., van der Laan-Luijkx, I. T., Krol, M. C., Spahni, R., Houweling, S., Gomez-Pelaez, A. J., van der Schoot, M., et al.: Development of CarbonTracker Europe-CH4—Part 2: Global Methane emission estimates and their evaluation for 2000–2012, *Geosci. Model Dev. Discuss, 2016.*

Turner, A. J., Frankenberg, C., Wennberg, P. O., and Jacob, D. J.: Ambiguity in the causes for decadal trends in atmospheric methane and hydroxyl, *Proceedings of the National Academy of Sciences, p. 201616020, 2017.*

---

## Author Response (AR1)

We responded to the reviewers' comments in detail in the two replies. Below is the new version of the manuscript, where changes with respect to the first version are marked.

[revised manuscript text omitted]

---

## Author Response (AR2)

We addressed all minor comments, as indicated in the marked PDF below.

We have finally decided to return to the former title: "Constraints and biases in a tropospheric two-box model of OH". Reviewer #2 suggested to add MCF and $CH_4$ to the title. We agreed that this would be more comprehensive, but the editor correctly stated that MCF should in that case be written out as methyl chloroform. Therefore, to prevent an overly long title we returned to the first version, which we think also covers the contents of the paper.

[revised manuscript text omitted]